# Electricity Generation from Renewable Energy Sources in Poland as a Part of Commitment to the Polish and EU Energy Policy

**Józef Paska [1],*, Tomasz Surma [2], Paweł Terlikowski [1] and Krzysztof Zagrajek [1]** 

[1]   Institute of Electrical Power Engineering, Warsaw University of Technology, Str. Koszykowa 75,
     00-662 Warsaw, Poland; pawel.terlikowski@ien.pw.edu.pl (P.T.); krzysztof.zagrajek@ien.pw.edu.pl (K.Z.)

[2]   CEZ Polska, Al. Jerozolimskie 63, 00-697 Warsaw, Poland; tomasz.surma@cezpolska.pl

*   Correspondence: jozef.paska@ien.pw.edu.pl

**Abstract:** The aim of this paper is to present the state of development of renewable energy sources (RES) in Poland in accordance with the obligations of European Union energy policy. The EU Member States are obliged to adopt different support mechanisms for the development of renewable energy sources, and in consequence to achieve their Directive's targets. Poland, being a Member State of the EU since 2004, has accepted a target of a 15% share of energy generated from renewable energy sources in final energy consumption, including 19.3% from renewable electricity until 2020. Due to the difficulties with target achievement, the authors found it reasonable to analyze the challenge of RES development in Poland. The article presents energy policy in the EU, as well as the review of measures implemented for renewable energy development. The current state of and perspectives on using renewable energy sources in Poland and in the EU are also depicted. In the article, the authors analyze the relation between reference prices at dedicated RES auctions in Poland and the levelized cost of electricity (LCOE). The paper also provides a description of the renewable energy sources' development in three areas: electricity, heat and transport biofuels. Its main content, though, refers to the generation of electricity from renewable energy sources.

**Keywords:** renewable energy sources; electricity generation; energy policy; electricity market; Poland; European Union

## 1. Introduction

Renewable energy sources (RES) are currently one of the pillars of energy development worldwide [1]. Their utilization contributes to the process of industry de-carbonization, thanks to much smaller amounts of emitted carbon dioxide ($CO_2$) [2]. However, it has to be remembered that their widespread use also raises many challenges for the electricity sector [3]. According to International Renewable Energy Agency (IRENA), the total installed capacity of renewable energy sources worldwide in 2019 was 2537 GW [4]. Solar photovoltaics (PV) and wind energy are mainstream options in the power sector, with an increasing number of countries generating more than 20% of their electricity with solar PV and wind. Almost half of this installed capacity—1118.969 GW—is located in Asia, including 758.626 GW in China. Europe has 573.266 GW of renewable capacity installed. The International Energy Agency (IEA) predicts the further increase in the new capacity of renewable sources to be approximately 160 GW in 2020, and 190 GW in 2021 [5].

As mentioned previously, Europe is investing heavily in renewable energy sources. This is related to the strategy that the European Union has adopted in the battle against excessive $CO_2$ emissions, at the same time boosting the transition to a low-carbon economy [6,7]. This has also been confirmed

by documents endorsed by the EU Parliament. The transition towards energy generation from low or zero emission sources in the EU was implemented in 1997 with the adoption of the White Paper [8]. Its introduction was aimed at impeding the import of coal, often of low quality, from outside the EU, and also at the introduction of the widely understood de-carbonization of the energy sector. Since 1997, in the scope of the promotion of renewable energy sources, the emphasis shifted from indicative targets (to be achieved in 2010) to legal targets. The aim is to define the basic legal framework for the transition from 2020's into 2030's perspectives, including a complex internal energy market mechanism [9]. Meanwhile, due to the dynamic development of renewable energy sources, new or revised support mechanisms for the construction of RES sources have been established.

Before 2010, the development of renewable energy was regulated by a legal framework which set indicative targets for each Member State. Directive 2001/77/EC of 27 September 2001 on the promotion of electricity produced from renewable energy sources in the internal electricity market [10] set national indicative targets to be achieved by the EU for the share of renewable energy in electricity production at 21% by 2010. Directive 2003/30/EC of 8 May 2003, on the promotion of the use of biofuels or other renewable fuels [11], set national indicative targets to be achieved by the EU for the share of renewable energy replacing petrol and diesel in transport at 5.75% by 2010.

Due to the rather small impact of the introduction of Directive 2003/30/EC, as well as the lack of commitment of Member States to the achievement of the targets for renewable energy sources, at the end of the first decade of the 21st century, the European Commission started working on a new Directive which defined the next step of the development of renewable energy sources in 2010–2020. It introduced provisions setting the mandatory targets and commitments of Member States. The directive was supported by the European Commission, EU Council and European Parliament [12].

In 2009, a new Directive (2009/28/EU) on the promotion of energy production from renewable energy sources was introduced [13]. Its provisions brought about a complete change of approach in the RES sector and in the functioning of the energy market [14]. The introduction of Directive 2009/28/EU unified the regulations and additionally introduced provisions concerning the production of heating and cooling from RES. Therefore, the new regulations allow the covering of all areas for RES development, electricity, heating and cooling and biofuels. Directive 2009/28/EU defines common objectives for the aforementioned areas, while offering the Member States flexibility in the implementation of individual solutions. However, it should be mentioned that the obligation of 10% use of biofuels in the transport sector was imposed. Figure 1 presents the objectives provided in the Directive for each of the Member States, as well as the output in the base year [13,15].

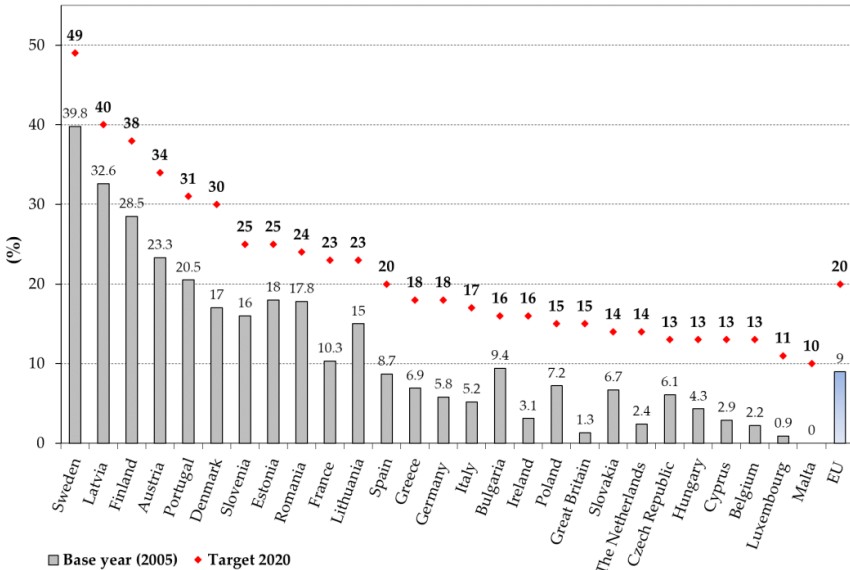

**Figure 1.** 2020's targets and share of renewable energy generation in the base year 2005 [13,15].

Now, in 2020, it is important to assess whether these goals have been achieved. Preliminary data shows that some Member States had already successfully met their targets ahead of schedule, while some may have difficulties meeting them before the end of 2020. The current situation of the COVID-19 pandemic affects the electricity market. With the introduction of the pandemic regime, in many power systems electricity consumption decreased. In March–April 2020, when the COVID virus has started to spread in the European Union, demand for electricity decreased by about 10–30%. This was mainly due to the production stoppage of many goods. As a result, this impact of the pandemic on the energy market and the behavior of energy consumers may affect the achievement of RES objectives by the end of 2020, as well as the future development of the energy sector and its transformation.

Table 1 presents targets and their realization in Member States. The EU as whole was, in 2018, on track to achieve the 2020 target, and countries like Bulgaria, Czech Republic, Denmark, Estonia, Greece, Croatia, Italy, Cyprus, Latvia, Lithuania, Finland and Sweden had achieved their targets before the expected date. In the following part of this article, the matter of RES target implementation by Poland is addressed in detail.

Furthermore, the European policy to promote the use of energy from renewable sources does not expire with the year 2020. In the 2014 Communication "A policy framework for climate and energy in the period from 2020 to 2030" [18], the European Commission started a debate on a new energy policy framework for the approach to 2030. Published in 2016, The European Commission's Communication "Clean Energy for All Europeans" is a document in which the European Commission indicated justifications for the proposed regulations in the field of the common electricity market, energy efficiency, renewable energy sources, security of energy supply and the governance principles of the Energy Union [19]. Over the next few years, discussions between the various parties, including the Member States, led to the adoption in 2018 of the package of final documents and regulations dedicated to specific areas and objectives for 2030. As a result, new targets for EU Energy Policy to be achieved by 2030 have been adopted [19], which follow the general de-carbonization policy [20]:

- At least 40% cuts in greenhouse gas emissions (from 1990 levels);
- At least 32% share for renewable energy;
- At least 32.5% improvement in energy efficiency.

The European Commission specified in these documents that in past years, investments related to the generation of energy from renewable sources represented over 85% of all investments in electricity generation. The introduction of new regulations, in the Commission's opinion, should consolidate this trend, and give a further push to the development of small, distributed sources using renewable resources, connected to the distribution network, including sources installed at the energy consumer level. Adopted in 2018, the revision of the Renewable Energy Sources Directive is intended to provide a new approach to the development of renewable sources after 2020 [21]. Therefore, it is an implementation of the provisions of the European Council, which set a target of a 32% share of energy being produced from renewable sources in total energy consumption in the EU by 2030. However, this time the Directive indicated the target for 2030 at the community level. It is a binding target at the EU level, but it does not translate into binding targets at the level of the Member States, as is the case in the current decade. In order to jointly achieve the target set for the EU, individual countries will declare their contribution to the development of integrated national energy and climate plans, which are part of the energy union's governance mechanism. These plans will be notified to the European Commission, and the Commission will have the option of ordering a review of these plans if it considers that the efforts and ambitions indicated by the Member State are not sufficient to achieve the Community target. National Energy and Climate Plans 2021–2030 of Member States were delivered to the European Union Commission in 2020.

At the same time, the Directive introduces market-based rules for the integration of renewable energy sources, based on equal and non-discriminatory access to the energy market and the energy transmission and distribution network. The new regulations remove the obstacle of priority access to

the network, and impose an obligation on operators of installations using renewable resources to bear production balancing costs. According to the principles of the Directive, support granted for the use of renewable energy resources should be distributed based on market principles, promoting competition between individual technologies.

**Table 1.** Share of renewable energy in final energy consumption (%) [16,17].

| Member State | 2005 | 2010 | 2014 | 2015 | 2016 | 2017 | 2018 | 2020 Target |
|---|---|---|---|---|---|---|---|---|
| Belgium | 2.3 | 5.6 | 8 | 7.9 | 8.7 | 9.1 | 9.4 | **13** |
| Bulgaria | 9.2 | 13.9 | 18 | 18.2 | 18.8 | 18.7 | 20.5 | **16** |
| Czech Republic | 7.1 | 10.5 | 15 | 15 | 14.9 | 14.8 | 15.1 | **13** |
| Denmark | 16.0 | 21.9 | 29.6 | 31 | 32.2 | 35.0 | 36.1 | **30** |
| Germany | 7.2 | 11.7 | 13.8 | 14.6 | 14.8 | 15.5 | 16.5 | **18** |
| Estonia | 17.4 | 24.6 | 26.3 | 28.6 | 28.8 | 29.1 | 30.0 | **25** |
| Ireland | 2.8 | 5.7 | 8.7 | 9.2 | 9.5 | 10.6 | 11.1 | **16** |
| Greece | 7.3 | 10.1 | 15.3 | 15.4 | 15.2 | 17.0 | 18.0 | **18** |
| Spain | 8.4 | 13.8 | 16.1 | 16.2 | 17.3 | 17.6 | 17.4 | **20** |
| France | 9.6 | 12.7 | 14.7 | 15.1 | 16 | 16.0 | 16.6 | **23** |
| Croatia | 23.7 | 25.1 | 27.8 | 29 | 28.3 | 27.3 | 28.0 | **20** |
| Italy | 7.5 | 13.0 | 17.1 | 17.5 | 17.4 | 18.3 | 17.8 | **17** |
| Cyprus | 3.1 | 6.2 | 8.9 | 9.4 | 9.3 | 10.5 | 13.9 | **13** |
| Latvia | 32.3 | 30.4 | 38.7 | 37.6 | 37.2 | 39.0 | 40.3 | **40** |
| Lithuania | 16.8 | 19.6 | 23.6 | 25.8 | 25.6 | 26.0 | 24.4 | **23** |
| Luxembourg | 1.4 | 2.9 | 4.5 | 5 | 5.4 | 6.3 | 9.1 | **11** |
| Hungary | 6.9 | 12.7 | 14.6 | 14.4 | 14.2 | 13.5 | 12.5 | **13** |
| Malta | 0.1 | 1.0 | 4.7 | 5 | 6 | 7.3 | 8.0 | **10** |
| Netherlands | 2.5 | 3.9 | 5.5 | 5.8 | 6 | 6.5 | 7.4 | **14** |
| Austria | 24.4 | 31.2 | 33 | 32.8 | 33.5 | 33.1 | 33.4 | **34** |
| Poland | 6.9 | 9.3 | 11.5 | 11.7 | 11.3 | 11.0 | 11.3 | **15** |
| Portugal | 19.5 | 24.2 | 27 | 28 | 28.5 | 30.6 | 30.3 | **31** |
| Romania | 17.6 | 22.8 | 24.8 | 24.8 | 25 | 24.5 | 23.9 | **24** |
| Slovenia | 16.0 | 20.4 | 21.5 | 21.9 | 21.3 | 21.1 | 21.1 | **25** |
| Slovak Republic | 6.4 | 9.1 | 11.7 | 12.9 | 12 | 11.5 | 11.9 | **14** |
| Finland | 28.8 | 32.4 | 38.7 | 39.2 | 38.7 | 40.9 | 41.2 | **38** |
| Sweden | 40.7 | 47.0 | 52.5 | 53.8 | 53.8 | 54.2 | 54.6 | **49** |
| United Kingdom | 1.1 | 3.8 | 7 | 8.5 | 9.3 | 9.7 | 11.0 | **15** |
| **EU** | **9.1** | **13.2** | **16.1** | **16.7** | **17** | **17.5** | **18.0** | **20** |

The Directive introduces an obligation to open support mechanisms for generation sources installed in other EU countries. In the years 2020–2025, the share for these foreign generation sources should be guaranteed at the level of 10% of new capacity being supported by support schemes, and from 2026 this share should increase to 15%. The purpose of the new approach is to stimulate the harmonization of support mechanisms and promote cross-border cooperation, including a common electricity market.

The new Directive puts more emphasis on the production of heating and cooling from renewable sources [22]. The Directive requires Member States to put in place mechanisms that will contribute

to an annual increase of 1 percentage point in the production of heating and cooling from renewable sources. Operators of heat distribution networks, especially so-called energy inefficient systems, will be exposed to greater competition. The Directive indicates the need to inform heat consumers about the share of heat derived from renewable energy sources and the efficiency of heating systems. For energy inefficient systems, Member States are to introduce the possibility of disconnecting individual heat consumers and enabling them to use individual heat supply solutions from renewable energy sources, such as heat pumps [23]. On the other hand, the draft directive requires non-discriminatory rules for the connection of heat sources using renewable resources to the district heating network [24,25].

Possible prospects for the development of renewable energy sources in Poland in the approach to 2030 under the regime of the new Directive are described in further parts of this article.

The organization of the paper is as follows: Section 2 provides information about implemented and future energy policy in Poland. Section 3 presents the current rules of the electricity market in Poland concerning the development of renewable energy sources. Section 4 presents an overview of the current status of RES in Poland. In Section 5, the RES development perspective is presented, with a special focus on a forecast regarding changes in the reference prices for RES auction and the levelized cost of energy (LCOE) in 2020–2025.

## 2. Renewable Energy in Energy Policy of Poland until 2030 and Draft of Energy Policy of Poland until 2040

### 2.1. Energy Policy of Poland until 2030

In 2009, the Polish Government adopted the Energy Policy of Poland until 2030, a strategy document wherein the main directions for the whole energy sector's development for the next 20 years were depicted [26,27]. This strategic document is still in force and has not yet been revised, despite the fact that the Polish government has prepared and published several drafts of a new energy policy and, more importantly, there have been changes in the conditions in which energy entities operate.

Information presented in the Polish Energy Policy until 2030 describes several tremendous challenges that the Polish energy sector has faced. These include, for instance, the increase in demand for energy and the insufficient and ageing power infrastructure, as well as a number of problems related to Poland's dependence on imported natural gas and oil. Considering the aforementioned problems and the obligations imposed by the EU in the field of environmental protection, especially the reduction in $CO_2$ emissions, it can be concluded that it is necessary for the authorities to take appropriate measures. However, it should be remembered that the implementation of the provisions of the Energy Policy may be verified by the general situation of the global energy sector and trends on energy markets. A new approach to the implementation of the provisions of the Energy Policy will be necessary in the case of significant fluctuations in the prices of energy resources, e.g., hard coal, but will also demand the analysis of failures of power systems in other countries, or legal amendments imposing even tighter $CO_2$ emission standards on Member States. As a Member State of the European Union, Poland actively participates in creating the Community energy policy, and it also implements its main objectives under the specific domestic conditions, taking into account the protection of the interests of customers, as well as the energy resources and technological conditions of energy generation and transmission.

In accordance with aforementioned, the primary directions of Polish energy policy are as follows:

- Energy efficiency improvement;
- Security of supply fuels and energy;
- Diversification of electricity generation by using nuclear energy;
- Renewable energy, including biofuels development;
- Competitive energy and fuel markets development;
- Influence of energy sector on environment limitation.

The presented directions of Poland's energy policy until 2030 are related to each other. Thanks to the implementation of measures aimed at improving energy efficiency, the increase in demand for fuels and energy can be limited. Each of the aforementioned elements may contribute to the reduction of pollution levels, which consequently reduces the impact of the energy sector on the environment. Moreover, the intensification of energy production via renewable energy sources and from biofuels will also have a positive impact on the reduction of emissions from the energy sector. Due to the use of innovative solutions in the field of electricity production and measures aimed at improving energy efficiency, the aforementioned directions of energy policy will make it possible to increase energy security, for example due to the reduction of imports of energy fuels, while ensuring sustainable development

The Policy presents demand for final energy—shown in Table 2—and demands for gross final energy from renewable energy sources by type—Table 3.

**Table 2.** Demand for final energy by carrier in Poland, Mtoe [15,26].

| Carriers | 2006 | 2010 | 2015 | 2020 | 2025 | 2030 |
|---|---|---|---|---|---|---|
| Coal | 12.3 | 10.9 | 10.1 | 10.3 | 10.4 | 10.5 |
| Oil products | 21.9 | 22.4 | 23.1 | 24.3 | 26.3 | 27.9 |
| Natural gas | 10.0 | 9.5 | 10.3 | 11.1 | 12.2 | 12.9 |
| Renewable energy | 4.2 | 4.6 | 5.0 | 5.9 | 6.2 | 6.7 |
| Electricity | 9.5 | 9.0 | 9.9 | 11.2 | 13.1 | 14.8 |
| District heat | 7.0 | 7.4 | 8.2 | 9.1 | 10.0 | 10.5 |
| Other fuels | 0.6 | 0.5 | 0.6 | 0.8 | 1.0 | 1.2 |
| **Total** | **65.5** | **64.4** | **67.3** | **72.7** | **79.3** | **84.4** |

**Table 3.** Demand for gross final energy from renewables in Poland, ktoe (adapted from [15]).

| | 2006 | 2010 | 2015 | 2020 | 2025 | 2030 |
|---|---|---|---|---|---|---|
| Electricity | 370.6 | 715.0 | 1516.1 | 2686.6 | 3256.3 | 3396.3 |
| *solid biomass* | 159.2 | 298.5 | 503.2 | 892.3 | 953.0 | 994.9 |
| *biogas* | 13.8 | 31.4 | 140.7 | 344.5 | 555.6 | 592.6 |
| *wind* | 22.0 | 174.0 | 631.9 | 1178.4 | 1470.0 | 1530.0 |
| *hydro* | 175.6 | 211.0 | 240.3 | 271.4 | 276.7 | 276.7 |
| *PV* | 0.0 | 0.0 | 0.0 | 0.1 | 1.1 | 2.1 |
| Heat | 4312.7 | 4481.7 | 5046.3 | 6255.9 | 7048.7 | 7618.4 |
| *solid biomass* | 4249.8 | 4315.1 | 4595.7 | 5405.9 | 5870.8 | 6333.2 |
| *biogas* | 27.1 | 72.2 | 256.5 | 503.1 | 750.0 | 800.0 |
| *geothermal* | 32.2 | 80.1 | 147.5 | 221.5 | 298.5 | 348.1 |
| *solar* | 3.6 | 14.2 | 46.7 | 125.4 | 129.4 | 137.1 |
| Transport biofuels | 96.9 | 549.0 | 884.1 | 1444.1 | 1632.6 | 1881.9 |
| *sugar and starch bioethanol* | 61.1 | 150.7 | 247.6 | 425.2 | 443.0 | 490.1 |
| *rape biodiesel* | 35.8 | 398.3 | 636.5 | 696.8 | 645.9 | 643.5 |
| *2nd generation of bioethanol* | 0.0 | 0.0 | 0.0 | 210.0 | 240.0 | 250.0 |
| *2nd generation of biodiesel* | 0.0 | 0.0 | 0.0 | 112.1 | 213.0 | 250.0 |
| *biohydrogen* | 0.0 | 0.0 | 0.0 | 0.0 | 90.8 | 248.3 |
| Total gross final energy from Renewable Energy Sources (RES) | 4780 | 5746 | 7447 | 10,387 | 11,938 | 12,897 |
| Gross final energy | 61,815 | 61,316 | 63,979 | 69,203 | 75,480 | 80,551 |
| Share of energy from renewables (%) | 7.7 | 9.4 | 11.6 | 15.0 | 15.8 | 16.0 |

According to the data presented in Table 3, the share of renewable energy sources will change significantly in the fuel mix of the Polish electricity generation sector by 2030. Figure 2 shows the current (year 2018) energy mix in the EU and Poland.

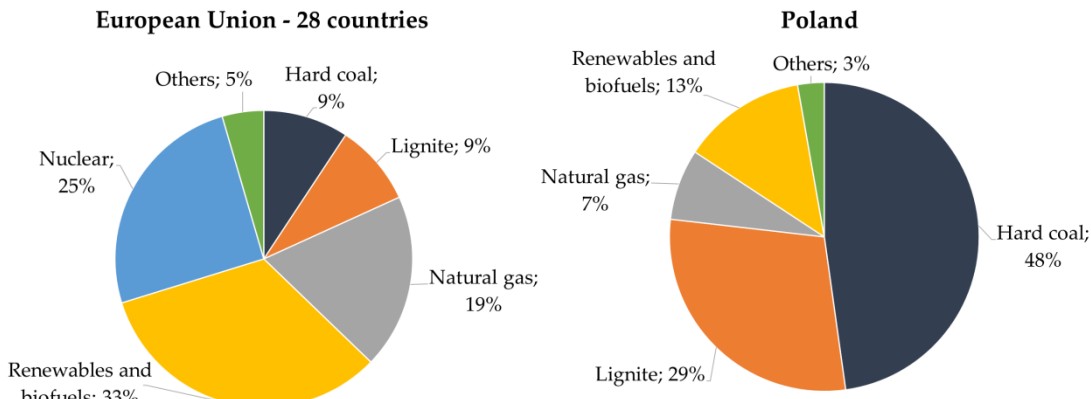

**Figure 2.** Structure of electricity generation in the EU and in Poland (year 2018).

Energy policy assumes that the use of renewable energy sources is one of the fundamental components of sustainable development in Poland. However, it should be taken into account that the dispatchability of RES power depends on both climatic conditions and the level of technological advancement (e.g., more efficient PV cells). According to the Polish Energy Policy until 2030, the highest potential for energy is shown by biomass (e.g., energy crops, firewood, forestry waste) and biogas (agricultural waste, sewage waste), as well as wind power (onshore and offshore) and hydropower [26]. This is mainly related to actual climate conditions, as well as the mechanisms of financing support introduced in Poland.

The implementation of support programs aimed at the development of electricity production from renewable energy sources is the foundation for the realization of the objectives set out in the Polish Energy Policy until 2030 [27]. As mentioned previously, increasing production from RES will reduce imports of energy fuels and thus allow for better energy independence. The production of electricity from RES also has a positive impact on the stimulation of the local economy, e.g., by the utilization of local energy resources. The more diversified the production technology, the more incentives there are for local businesses. It should be remembered that energy production from RES often takes place in distributed units, so that many Polish regions, especially agricultural ones, may gain some economic benefits. Moreover, the local security of electricity supply will be increased, thanks to the fact that energy production will take place in sources located close to the group of end users, and losses in energy transmission will be limited. The production of energy from renewable sources also means the reduction of the carbon footprint, which is a positive environmental effect. The key to the effective use of renewable energy sources is to maximize the extraction of primary resources. One example of such activities is the use of biogas, which is obtained from landfills or sewage plants.

Considering also the high availability of energy from biogas sources, a rapid increase in energy production from such renewable energy sources is expected. Therefore, measures that ensure the stability of the power system's operation using advanced techniques are becoming even more important than ever [9,28].

The key targets of the energy policy in the area of renewable energy are as follows:

- Increasing the use of renewable energy sources in final energy consumption up to at least 15% in 2020, and continued development in subsequent years;
- Raising the share of biofuels in the transport fuel market up to 10% by 2020, and expanding the use of second generation biofuels;

- Preserving forests against over-exploitation for the production of biomass and the sustainable exploitation of agricultural areas for the production of energy from renewable sources, including biofuels, in order to avoid any competition between renewable energy production and agriculture, as well as to maintain environmental diversity;
- Utilization of existing government-owned weirs for hydropower production;
- Enhancing diversification among supply sources and establishing favorable circumstances for distributed electricity generation, which are both based on locally available resources.

Today, many elements of energy policy require revision and evaluation. It is necessary to adapt and to set new directions for action in light of new circumstances and challenges for the energy sector. Many of the measures already enshrined in 2009 have been implemented, but many of them have also already become outdated with the development of the energy market and the new conditions that arose at that time.

### 2.2. Draft of Energy Policy of Poland until 2040

On 23 November 2018, the former Ministry of Energy, which was functioning within the administration's structures at that time, published a draft document "Energy Policy of Poland until 2040" [29].

The project assumes the implementation of a new energy policy from eight courses of action in the energy sector [29]:

1. Optimal use of own energy resources;
2. Development of electricity generation and network infrastructure;
3. Diversification of natural gas and oil supply, and development of network infrastructure;
4. Development of the energy markets;
5. Implementation of nuclear energy in Poland;
6. Development of renewable energy sources;
7. Development of district heating and cogeneration;
8. Improvement of energy efficiency.

The draft of the Energy Policy until 2040 assumes Poland will strive to cover the demand for energy with its internal resources. Domestic coal resources will remain the main source of Poland's energy security and the core of its energy balance. The use of coal by the power plants will continue to be stable, but the share of coal in the structure of energy consumption will decrease—to about 60% in 2030. At the same time, an increase in energy consumption is expected, which will be covered by sources other than conventional coal capacity (the source most commonly used today). Investments in new coal-based power plants undertaken after 2025 will be based on highly efficient cogeneration units, or other technology meeting the emission standard of at least 450 kg $CO_2$ per MWh of generated energy.

In terms of the structure of energy carriers, it is planned to maintain the significant role of coal, but due to the expected increase in energy demand, the necessity of reducing $CO_2$ emissions, as well as the principle of rationality and the implementation of new technologies, there will gradually be a drop in the percentage share of coal in the structure of electricity generation. The above-mentioned trends will continue into the next decade, i.e., until 2040. The diversification of energy carriers gradually increases the share of RES. It will also be important to increase the use of natural gas. Access to diversified natural gas sources and the availability of easily accessed gas as a commodity, at a price acceptable to the end user, will allow the wider use of this resource for heating and electricity generation in the national economy. This may result in reducing the carbonization of the economy, while at the same time being an effective tool for clean air, as stated in the policy act.

Renewable energy sources will play an increasing role in power systems. Their level in the structure of national electricity consumption may amount to about 27%. However, achieving such a share in the electricity consumption development of energy storage is necessary, as is the expansion of gas-fueled units as balancing power units.

The main instrument to reduce emissions from the energy sector will be the implementation of nuclear energy in 2033. It has been assumed that by 2043, six nuclear units, with a total capacity of 6–9 GW, will be built. This means that in 2040 the share of this energy in electricity generation could be about 10%.

Nowadays, the implemented capacity market has a significant impact on the shape of the sector and on power adequacy. The draft of the Energy Policy points out that the Minister of Energy, based on market analysis and forecasts of market growth until 2024, will decide whether the continuation of the capacity market is required.

In Energy Policy, and during official meetings with the European Commission, Poland has declared it will achieve a 21% share of renewables in final energy consumption by 2030. The government has suggested that this level of renewable development is in line with the requirements of the 15% 2020 target. Photovoltaic installations and offshore wind power plants are presented as the most promising routes for development in Poland in the next decade.

Currently, more than 2 GW of PV is installed, and every month we have observed dynamic development in the new installations. The Energy Policy assumed that the development of photovoltaics will take place from 2022, which relates to the reduction of costs of this technology. The Policy paper assumes the installation of 1 GW capacity of PV per year. It is estimated that 20 GW of PV installed capacity could be reached by 2040, with an electricity production of about 20 TWh.

Offshore wind energy is set for implementation in 2027. The start of investment in this technology is determined by the completion of works on the reinforcement of the transmission grid in the northern part of the country. It is expected that the first investment will give about 1.2 GW of offshore capacity. In the second stage, thanks to the further development based on experience from existing projects, about 10 GW of wind power may be installed in the period of 2035 to 2040. This will lead to the generation of about 41 TWh of electricity from offshore wind energy. For these sources, the capacity factor may exceed 45% in 2040.

Remarkably, in the case of onshore wind power, the Ministry intends to phase out existing wind power capacity from the second half of this decade. Several new technologies are scheduled to be installed. It was assumed that the development of hydro energy in the long-term horizon may be positively influenced by the development of inland waterways and the revitalization of water dams. Regardless, the policy does not expect a huge development of hydro power plants in Poland.

The capacity of other RES sources, such as biomass and biogas, according to the Policy's forecasts, is also growing slightly. The new capacity derived from these sources in the years 2035–2040 will amount to 4.3 GW. In particular, the development of cogeneration, including the conversion of heat plants, the optimization of the existing systems and their reconstruction, was considered. It was assumed that it will be possible to rebuild cogeneration capacity by building new cogeneration units in coal, gas or biomass technology, with a heat output equivalent to that of cogeneration units. The draft of policy aims to maintain support for the electricity generated in high-efficiency cogeneration, as well as the increase in its use. It was assumed that it will be possible to restore the cogeneration capacity by building new cogeneration units in coal, gas or biomass technology. An increase in the use of RES in district heating was also envisaged, which will take place mainly using local renewable energy resources, i.e., biomass, biogas or geothermal sources. The draft policy also emphasizes the increased use of waste in district heating systems.

The following indicators have been taken as the global measures of achievement of the objectives of "the Polish Energy Policy until 2040" [29]:

- 60% share of coal in electricity generation in 2030;
- 21% of RES in gross final energy consumption in 2030;
- implementation of nuclear power in 2033;
- improvement of energy efficiency by 23% by 2030, compared to forecasts from 2007;
- reduction of $CO_2$ emissions by 30% by 2030 compared to 1990.

## 3. Rules for Renewable Energy Source Development in Poland

*3.1. General Information*

Under Directive 2009/28/EC of the European Parliament and the Council of April 23rd 2009 on the promotion of the use of energy from renewable sources, Poland is obliged to derive a minimum 15% share of energy from renewable sources in its gross final energy consumption by 2020. In addition, in the transport sector, Poland is committed to achieving a 10% share of biofuels use by 2020. In 2010, the Polish Government presented the "National Renewable Energy Action Plan with forecast and strategy for development of renewables in Poland till 2020" [30]. The overall national target of a 15% share of RES in gross final energy consumption, in accordance with Directive 2009/28/EU, corresponds to a 10,380 ktoe consumption of renewable energy in 2020, compared to the expected 69,200 ktoe of total energy consumption in Poland in 2020. The measures were identified by which Poland is implementing the 2020 target, broken down into individual energy consumption areas. The Plan also sets a national target for the share of energy derived from renewable sources in gross final energy consumption of 15.85% (considering the surplus for the cooperation mechanism under the Directive). Partial goals have been defined for:

- district heating and cooling—17.05%;
- electricity generation—19.13%;
- transport—11.36%.

The trajectories of the goals in each sector are presented in Figure 3.

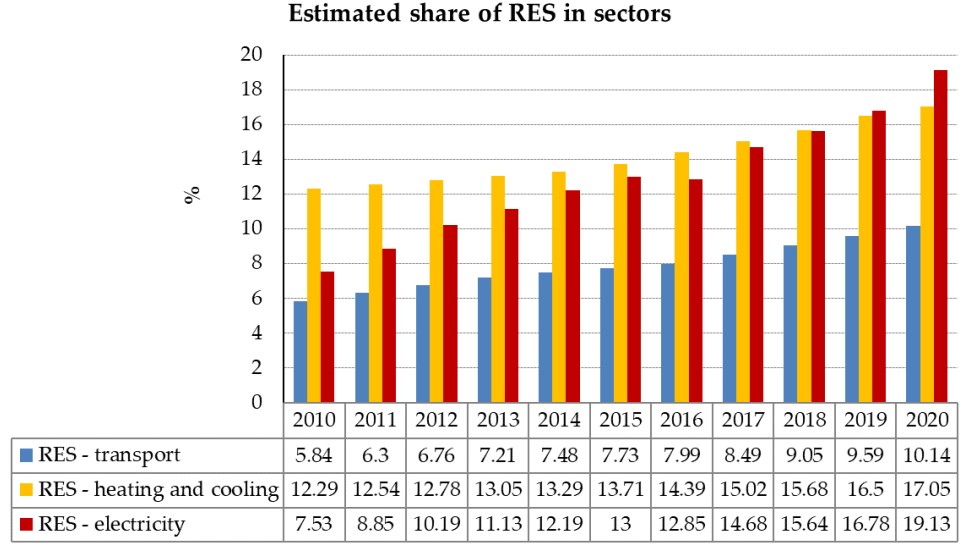

**Figure 3.** Share of renewable energy generation in different sectors according to the National Renewable Energy Action Plan [30].

The Action Plan presented biomass, biogas and wind energy as the most promising means to achieve the goal. Hydropower, due to Poland's potential, is mainly limited to plants with a capacity below 1 MW. The Plan also did not present the possibility of generating electricity from geothermal energy, and also only outlined a very limited degree of photovoltaics use, due to the significant costs of these technologies at that time.

Table 4 presents the total estimated contribution (installed capacity, gross electricity production) of each RES technology to the achievement of binding targets for 2020, and an indicative course for the share of RES energy in power generation between 2010 and 2020.

**Table 4.** Data from National Action Plan for renewable energy sources towards achieving the 2020 target [30].

| | 2005 | 2010 | 2015 | 2016 | 2017 | 2018 | 2019 | 2020 |
|---|---|---|---|---|---|---|---|---|
| **Installed Capacity (MW)** | | | | | | | | |
| hydro energy | 915 | 952 | 1002 | 1012 | 1022 | 1032 | 1042 | 1152 |
| wind energy | 121 | 1100 | 3540 | 4060 | 4580 | 5100 | 5620 | 6650 |
| solar PV | 0 | 0 | 2 | 2 | 3 | 3 | 3 | 3 |
| biomass | 286 | 380 | 1530 | 1630 | 1780 | 1930 | 2230 | 2530 |
| **Total** | **1322** | **2432** | **6074** | **6704** | **7385** | **8065** | **8895** | **10,335** |
| **Gross Electricity Generation (GWh)** | | | | | | | | |
| hydro energy | 2201 | 2279 | 2439 | 2471 | 2503 | 2535 | 2567 | 2969 |
| wind energy | 136 | 2310 | 7541 | 8784 | 9860 | 11,210 | 12,315 | 15,210 |
| Solar PV | 0 | 2 | 2 | 2 | 3 | 3 | 3 | 3 |
| biomass | 1451 | 6028 | 9893 | 10,348 | 11,008 | 11,668 | 12,943 | 14,218 |
| **Total** | **3788** | **10,619** | **19,875** | **21,605** | **23,374** | **25,416** | **27,828** | **32,400** |

In order to meet the EU target and to ensure a continuous increase in installed capacity derived from renewable energy sources, the following support mechanisms have been introduced in Poland:

- a system of green certificates;
- an auction support system;
- a feed-in-tariff scheme—dedicated to installations using biogas, biomass and hydropower, with an installed capacity not exceeding 500 kW;
- a feed-in-premium scheme—dedicated to installations with installed capacity in the range between 500 kW and 1 MW;
- an obligation to purchase 'green energy' from an installation with a total installed capacity of <500 kW by the obliged sellers with a price corresponding to the average electricity sales price on the competitive market in the previous quarter, as announced by the President of the Energy Regulatory Office (ERO);
- a discount scheme, dedicated to micro-installations with an installed capacity up to 50 kW, so-called prosumers of energy.

*3.2. Green Certificates Scheme*

The support system based on the formula of green certificates was introduced into Polish law for the first time in 2004, after securing European Union membership [31]. The system generates additional revenues for producers of electricity from renewable energy sources via the sale of certificates, which are above the revenues derived from the sale of electricity.

The Energy Act introduced an obligation to obtain, and submit for redemption to the President of the Energy Regulatory Office, certificates of origin for electricity generated from a renewable energy source—so-called green certificates—or to pay a substitution fee.

Obligations are imposed on:

- energy companies performing activities in electricity generation or the trading and selling of this energy to end users who are not industrial customers;
- industrial customers who consumed yearly no less than 100 GWh of electricity, and the cost of which was no less than 3% of its production value;

- end users of electricity, other than industrial users, who are members of a commodity exchange or of a market organized by the operator of the regulated market with respect to the transactions concluded in their own name;
- commodity brokerage houses or brokerage houses purchasing electricity for end customers other than industrial customers.

These provisions and this obligation for green certificates were transferred from the completed support system to a dedicated Act on renewable energy sources [32]. A general rule of the Act on renewable energy sources established for all RES power plants a maximum 15 years of support from the first day of electricity generation, when the certificate of origin was issued, which means the day when the support system was applied for the particular RES installation for the first time.

Additionally, the dedicated ordinance specifies the quantitative share of the sum of electricity resulting from green certificates (which entities are obliged to submit for redemption or else pay a substitution fee) in the total annual sales of electricity by that entity to end users. In this way, the Act defines entities obliged to generate demand for certificates, and the secondary regulation determines the demand for certificates. The scope of obligation, defined in the regulation, determines the percentage share of electricity produced from RES sold to the final consumer, and is shown in Figure 4.

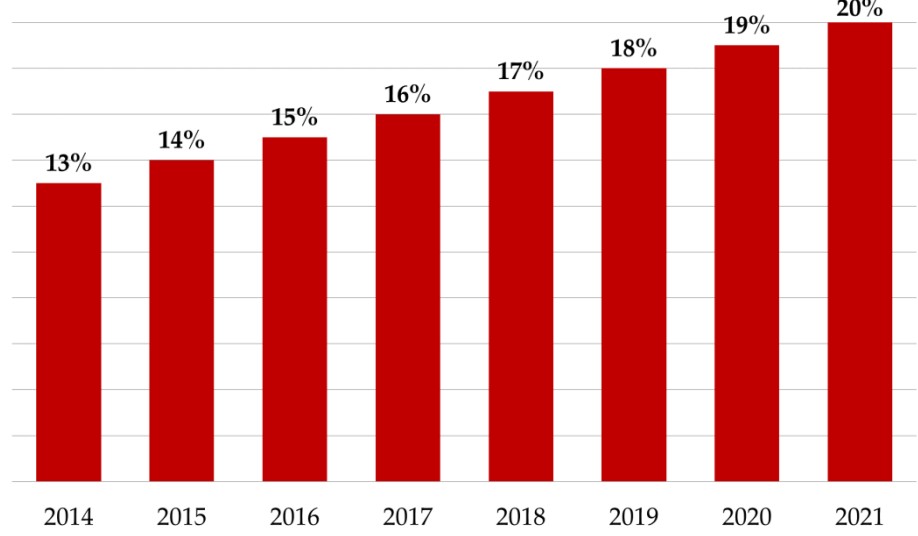

**Figure 4.** The yearly targets of renewable electricity generation in Poland.

At present, there is a significant oversupply on the green certificates market, i.e., for several periods, electricity production in this system exceeded the demand for certificates. This has led to a significant accumulation of certificates in the register. The oversupply is not something new; it existed in the system and is also responsible for the liquidity of this market. However, the oversupply is an essential element which determines certificate prices. Significant oversupply was created in 2014–2018, when, after the beginning of the decade, the system was left to its own devices and the obligation to purchase certificates at the same level was maintained without any government intervention. The data presented in Figure 5 shows that in the years when oversupply was accumulating, prices started to fall sharply, to a critical minimum in 2017.

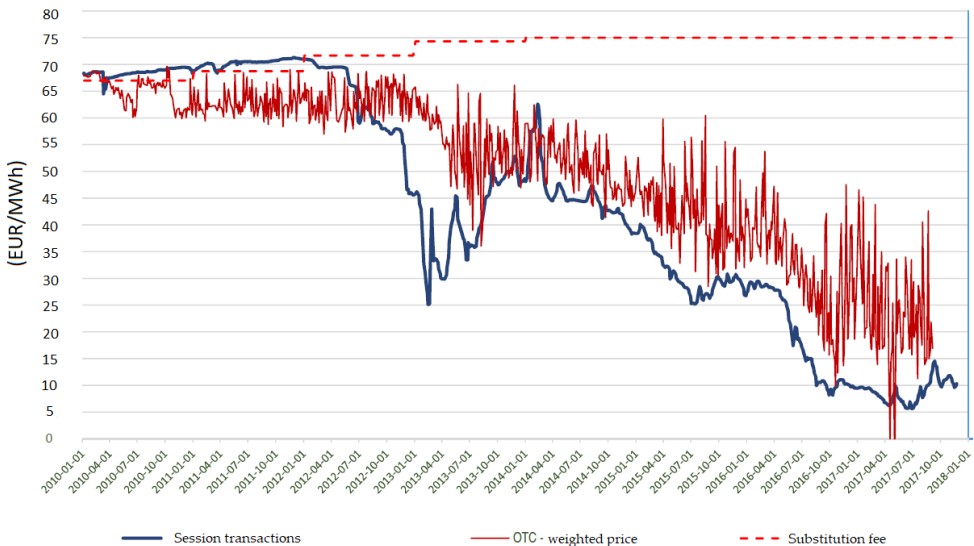

**Figure 5.** Prices of green certificates in period 2010–2018 [33].

However, the RES support system generates costs for final customers [34,35]. In the Polish support system, the cost of green certificates is distributed directly to end users and is included in their energy invoices. Table 5 presents the cost of green certificates related to energy production in 2006–2019. The calculations are based on the amount of RES electricity produced and the price of the substitution fee (as the maximum level of support). The cost of the RES support system for renewable electricity per unit of generated energy in Poland (EUR 71.72/MWh in years 2014 and 2015) is slightly below the European average (EUR 110/MWh in the year 2015) [36].

**Table 5.** Cost of support for renewable electricity generation in Poland, EUR $10^6$.

| | 2006 | 2007 | 2008 | 2009 | 2010 | 2011 | 2012 | 2013 | 2014 | 2015 | 2016 | 2017 | 2018 | 2019 [1] |
|---|---|---|---|---|---|---|---|---|---|---|---|---|---|---|
| Cost of green certificates | 236 | 295 | 375 | 518 | 679 | 799 | 1058 | 807 | 733 | 567 | 384 | 182 | 512 | 721 |

[1] For 2019, due to uncompleted data, this is only an estimation.

### 3.3. Auction Scheme

The Act on renewable energy sources redesigned the approach to new investments in renewable energy sources, as commissioned on 1 January 2016 [32]. To optimize the support level, the Government decided to apply an auction support scheme for new installations. The result of the auction system is a contract for the differences in prices of electricity generated via renewables. This auction scheme is a new model of support system, and results from the EU recommendation on State Aid Guidelines for energy and environment in the period 2014–2020. In the auction system, investors compete for support among each other.

The subject of the auction is the volume of electricity generated in a period of 15 years, and the auction's criterion is the price of generated electricity, PLN/MWh. Auction organization should be planned at minimum once per year. Before the auction, the Government will announce the expected amount of electricity in the auction, and the maximum price in tenders. Furthermore, the auction budget in PLN is presented in advance. For the result of a "blind auction", the President of the Regulatory Office presents to the public the winners of the auction and the benefiters of the new support scheme. Winning offers will apply to investors for a period of 15 years, with yearly updates in accordance with the inflation rate [32].

Auctions are arranged in separate pools for the following types of RES units:

(1)  existing RES power plants;

(2)  new RES power plants, which are commissioned after 1 January 2016, and with a capacity threshold of 1 MW.

The system has been implemented so far. In 2016–2019, as a result of auctions for new installations, the total amount of contracted energy is about 152.7 TWh. Its value is PLN 36.78 billion, which amounts to EUR 8.55 billion. The execution of the contracted winning projects will allow the construction of new RES power plants with a total capacity of 5037 MW, in the following generation technologies:

- 3151 MW of onshore wind energy with capacity above 1 MW;
- 328 MW of PV with capacity above 1 MW;
- 1472 MW of PV with capacity below 1 MW;
- 12 MW of biomass, in a dedicated biomass combustion plant with capacity >1 MW;
- 8.5 MW of biomass, in a dedicated biomass combustion plant and with high-efficiency cogeneration, with power below 50 MW;
- 16 MW of small hydropower with capacity below 1 MW;
- 7 MW of hydropower with capacity above 1 MW;
- 7 MW of agricultural biogas with capacity above 1 MW;
- 36 MW of agricultural biogas with capacity below 1 MW.

The photovoltaic projects that won the auction in 2016—about 116 MW—have already had to be put into operation. According to the law, those projects are obliged to start energy production within 24 months. A significant part of these projects, especially those that won the auctions conducted in 2017–2019, is still under construction. Considering that the main volume was distributed via auctions held in 2019, some of these projects will not be built by the end of 2020, and thus will not contribute to the RES target for that year. It is worth mentioning that the Polish auctions for wind power plants in 2019 were the largest in the European Union that year. In an optimal scenario, the projects that won the RES auctions conducted between 2017 and 2019 could generate 2.2 GW of new capacity by the end of 2020.

Table 6 presents the prices of the winning projects in the RES auctions 2016–2019.

**Table 6.** Prices in auctions in period 2016–2019.

| Capacity in Auction | Type of Power Plant | Min. Price | | Max. Price | |
|---|---|---|---|---|---|
| | | (PLN/MWh) | (EUR/MWh) | (PLN/MWh) | (EUR/MWh) |
| >1 MW | wind energy onshore and PV | 171.88 | 39.97 | 283.05 | 65.83 |
| ≤1 MW | PV | 271.25 | 63.08 | 386.90 | 89.98 |
| ≤1 MW | hydro power plants | 418.70 | 97.37 | 479.99 | 111.63 |
| >1 MW | biomass | 400.00 | 93.02 | 400.00 | 93.02 |
| >1 MW | Combined Heat and Power (CHP) biomass with capacity below 50 MW | 400.00 | 93.02 | 400.00 | 93.02 |
| >1 MW | agricultural biogas | 496.00 | 115.35 | 517.00 | 120.23 |
| ≤1 MW | agricultural biogas | 538.86 | 125.32 | 569.69 | 132.49 |

The low price achieved by onshore wind indicates its cost advantage over other power generation technologies. The average price from the bids submitted in the 2019 auction for onshore wind power plants (amounting to PLN 208/MWh or EUR 48.37/MWh) was once again lower than the TGeBase price (arithmetic mean of the weighted average hourly prices of a given delivery day from 00:00 to

24:00, calculated on the basis of all hourly contracts and weekends). The price of electricity on the Polish Power Exchange in forward contracts for 2021 (BASE) was, in 2019, about PLN 252/MWh (EUR 58.60/MWh). The results of these auctions proved that the technology of wind power plants in Polish conditions has become a commercial technology, which allows the securing of its revenues from the sale of electricity only on the competitive energy market. It can also be expected that these operators have secured in auction the sale of energy that is only part of the volume that can be produced, and the rest is sold under commercial energy market rules. On 31st December 2019, the Council of Ministers adopted a draft regulation on the maximum quantity and the value of electricity from renewable energy sources that may be auctioned in 2020. According to the ordinance, there is a plan to sell electricity generated by new installations, thanks to which about 2–3 GW of capacity can be commissioned. The regulation provides auctions for about:

- 1000 MW of onshore wind energy with capacity above 1 MW;
- 780 MW of small PV below 1 MW;
- 95 MW of biomass, in a dedicated biomass combustion plant, and biogas from landfills and from sewage treatment plants with capacity above 1 MW;
- 30 MW of hydro power, including 10 MW of small hydro below 1 MW;
- 15 MW of agricultural biogas with capacity below 1 MW.

The ordinance also introduced the possibility of transferring from the green certificates scheme into auction one of the agricultural biogas plants (with a total installed electrical capacity above 1 MW) with a total volume of about 20 MW [32].

A summary of the auction conducted in the period 2016–2019 is presented in Section 4 of this paper.

### 3.4. Feed-in-Tariff Scheme and Feed-in-Premium Scheme

The Act on Renewable Energy Sources implemented dedicated rules for small installations [32,37]. Among others, it guaranteed that the price of electricity was implemented in the form of either feed-in tariffs for small-scale and micro-scale biogas and hydro installations (with a capacity lower than 500 kW), or a right to recover any negative balance in the form of feed-in premium tariffs for mid-scale biogas and hydro installations (with a capacity higher than 500 kW but lower than 1 MW). The system is recruited and declarations of willingness to use the system are accepted on a continuous basis (as opposed to the auction system in which the President of ERO announces the relevant auction sessions). The generator submits to the Energy Regulatory Office a declaration of willingness to sell energy within the tariff system. The Act gives a tool to the Council of Ministers to determine, by way of an ordinance, the maximum installed capacity of particular types of renewable energy source installations in the tariffs scheme. The regulation for the following year may be issued until 31st October of a given calendar year. The tariff purchase price is 90% of the reference price for a given type of installation, and is indexed annually to the total average annual consumer price index of the previous calendar year.

### 3.5. Discount Scheme, Dedicated for Prosumers

The Act on renewable energy sources defined a dedicated scheme for micro-installations, addressed to end users of energy [32]. According to the Act, a prosumer is an end-user purchasing electricity pursuant to the comprehensive agreement, producing electricity exclusively from the renewable sources of energy in a micro-installation with the purpose of consuming it themselves, unrelated to executed economic activity. At the same time, the Act defined a micro-installation as a facility of renewable energy production with a total electrical installed capacity no higher than 40 kW, connected to the grid at a nominal voltage lower than 110 kV, or having an available thermal capacity in cogeneration no higher than 120 kW.

For such solution systems there is a dedicated discount scheme. The seller settles the volume of electricity input by the prosumer to the grid in relation to the volume of electricity derived from the network in the ratio 1 to 0.7, excluding micro-installations of total installed electric power no higher

than 10 kW, for which this ratio is 1 to 0.8. The discount also refers to micro-installations, for the construction of which public aid was granted (e.g., in the form of grants).

The seller makes a settlement on obtaining metering data from the distribution system operator, transferred by this operator in such way that the volume of energy input and energy consumed by the prosumer is settled in accordance with the earlier balancing of the volume of energy from all phases of the three-phase micro-installations.

The support mechanisms presented above exist together, with the general rule that the use of one of them excludes participation in another. The green certificate system has been in operation for the longest time, and has so far brought with it the largest amount of installed renewable capacity. The next chapter presents the current state of the development of renewable energy, and prospects for further development after 2020.

## 4. Current Status of RES in Poland

### 4.1. General Information

According to Eurostat data, the share of energy from renewable sources in the gross final energy consumption of 2017 was 10.96%, while in 2018 it was about 11.28% [38]. This is the official indicator used to monitor the 2020 target set in Directive 2009/28/EC on the promotion of the use of energy from renewable sources. Gross final energy consumption is defined in the Renewable Energy Directive 2009/28/EC as energy goods supplied for energy purposes to industry, transport, households, services (including public services), agriculture, forestry and fisheries, including the consumption of electricity and heat by the energy industry for the production of electricity and heat, and also including losses of electricity and heat during distribution and transmission. The share of energy derived from RES in the gross final consumption of energy in 2018 did not reach the planned ratio (14.09%) in the National Action Plan [30]. Moreover, it was lower than the shares in 2015 and 2016. Taking into account the data for 2018, achieving the RES target for 2020 requires an increase in the share of energy derived from renewable sources in final energy consumption by over 4.0 percentage points. The anticipated stagnation in electricity allows for the assumption that Poland may not achieve the mandatory minimum share of energy derived from RES in its total gross energy consumption by 2020. Table 7 presents the share of energy derived from renewable sources in the gross final consumption of energy [39], and Figure 6 presents the progress towards the 2020 renewable energy source target for Poland [38,39].

**Table 7.** Share of energy derived from renewable sources in gross final consumption of energy, in (%) [39].

|  | 2005 | 2010 | 2015 | 2016 | 2017 | 2018 |
|---|---|---|---|---|---|---|
| Share of energy from renewable sources (%) | 6.895 | 9.253 | 11.743 | 11.267 | 10.964 | 11.284 |

Furthermore, in none of the sectors (heating and cooling, electricity and transport) did the share of energy derived from renewable sources in the gross final consumption of energy in 2018 exceed the level assumed in the National Action Plan [30]. In the heating and cooling sector, the share of energy derived from renewable sources was 14.79% (compared to 15.68%, as planned). In the electricity sector, it amounted to 13.03% vs. the 15.68% planned in the Action Plan, and in the transport sector, it was 5.63% vs. the 10.08% planned to be achieved in 2018. Unfortunately, in 2019, there was again no significant development in the growth of energy generation from renewable sources. Based on that fact, the possibility of achieving the target by 2020 is questionable.

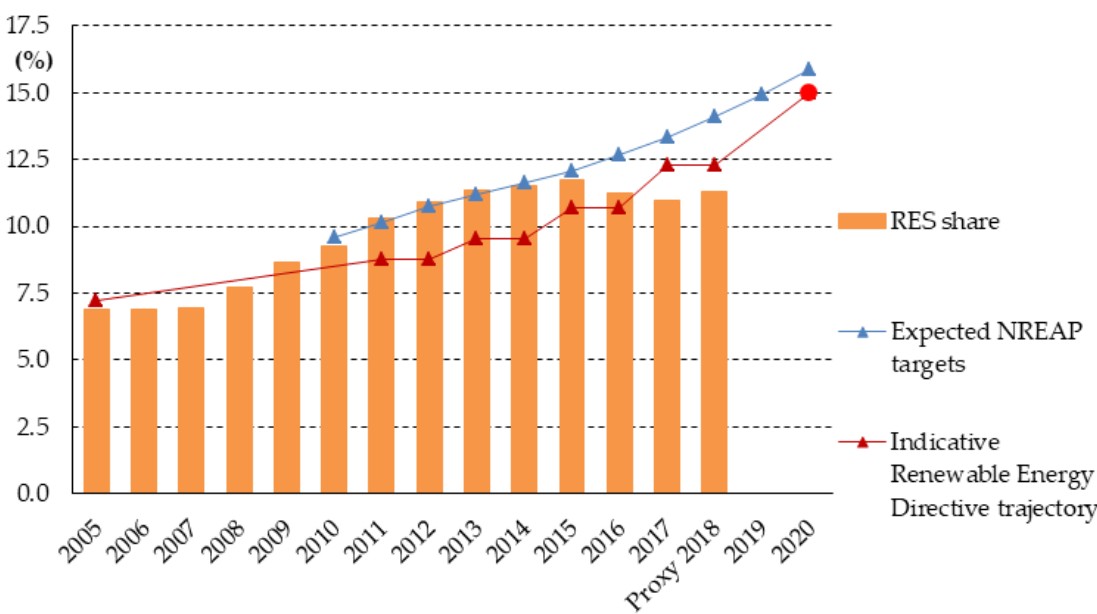

**Figure 6.** Progress towards renewable energy target, in (%) [38,39].

Figure 7 presents the share of renewable energy in the gross final energy consumptions of the transport, heating and cooling and electricity sectors in Poland so far [40].

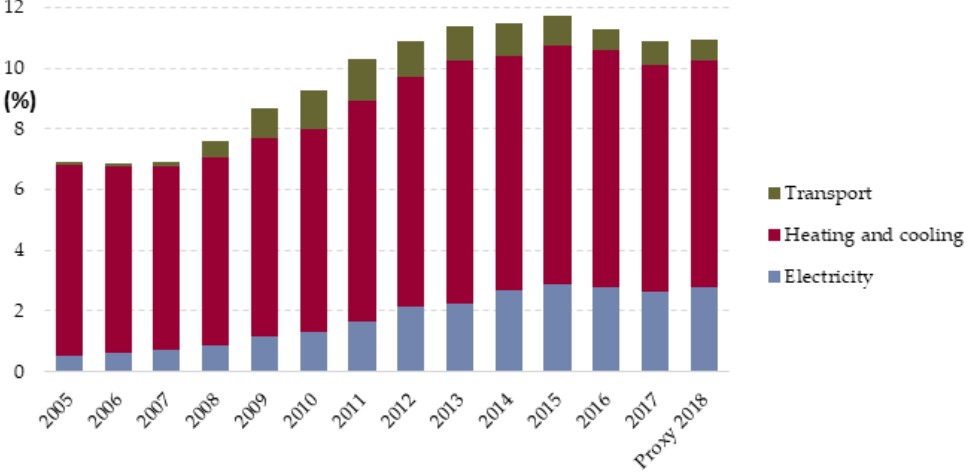

**Figure 7.** Share of renewable energy in gross final energy consumption in Poland [40].

According to the Energy Regulatory Office data, as of 31 December 2019, renewable energy source installations with a total of 9106 MW of production were in operation. They all obtained a license to produce electricity, and entered into the register of small installations or the register of agricultural biogas and micro-installations. A significant part of these capacities was developed and is currently operating under the green certificate regime. Table 8 presents RES installed capacity in Poland.

At the same time, in recent months we have observed a dynamic growth in the installed capacity of PV sources. These are the installations that won the auctions in 2016–2019, but also micro-installations implemented in the prosumer system. According to PSE (Polskie Sieci Elektroenergetyczne) data, as of 1 July 2020, the installed capacity of photovoltaics was 2108.9 MW. This means a 175% year-to-year increase in this technology.

**Table 8.** RES-E installed capacity in Poland, in (MW) [40].

| RES | 2005 | 2010 | 2015 | 2016 | 2017 | 2018 | 2019 |
|---|---|---|---|---|---|---|---|
| biogas | 32.0 | 82.9 | 212.5 | 234.0 | 235.4 | 237.6 | 245.4 |
| biomass | 189.8 | 356.2 | 1122.7 | 1281.1 | 1362.0 | 1362.9 | 1492.9 |
| photovoltaics | 0.0 | 0.0 | 71.0 | 99.1 | 103.9 | 147.0 | 477.7 |
| onshore wind | 83.3 | 1180.3 | 4582.0 | 5807.4 | 5848.7 | 5864.4 | 5917.2 |
| hydro | 852.5 | 937.0 | 981.8 | 994.0 | 988.4 | 981.5 | 973.1 |
| **Total** | **1157.5** | **2556.4** | **6970.0** | **8415.5** | **8538.3** | **8593.4** | **9106.3** |

At the same time, renewable electricity generation increased from 2.8 TWh in 2004 to almost 24 TWh in 2019. Table 9 presents electricity generation from RES in Poland in different technologies.

**Table 9.** Electricity generation from renewable energy sources in Poland in 2005–2019, in (GWh) [40].

| RES | 2005 | 2010 | 2015 | 2016 | 2017 | 2018 | 2019 |
|---|---|---|---|---|---|---|---|
| hydro | 2143.5 | 2349.0 | 2353.4 | 2347.6 | 2326.9 | 2335.3 | n/a |
| onshore wind | 203.3 | 1700.3 | 9687.5 | 12,040.9 | 13,570.8 | 13,655.9 | n/a |
| photovoltaics | 0.0 | 0.0 | 56.6 | 123.9 | 165.5 | 300.5 | n/a |
| biomass | 1399.9 | 5905.2 | 9026.6 | 6912.7 | 5308.6 | 5333.2 | n/a |
| biogas | 111.3 | 399.3 | 906.4 | 1040.3 | 1177.1 | 1212.6 | n/a |
| **Total** | **3857.9** | **10,353.8** | **22,030.6** | **22,465.4** | **22,548.8** | **22,837.5** | **23,937.5** |

One may notice that after implementation of the support scheme based on green certificates in 2004, electricity generation increased, especially from wind, biomass and biogas. Hydro seems to be in a stable position, and only the development of small units, below 1 MW, is observed. Furthermore, in 2004 the Ministry of Economy accepted the co-firing of biomass with fossil fuels as renewable (only the part of the biomass in the total fuel amount). That is why co-firing was quite popular in Poland in the period 2004–2016 (used in 44 system power plants' and Combined Heat and Power (CHP) units, reaching almost 10 TWh of generation). However, after this "golden age" period the government decided to withdraw this technology in favor of dedicated biomass boilers supported by the auction scheme. Anyway, this approach, new in the policy of renewable energy and biomass utilization, has not yet delivered the expected results. In the power sector, there has been a collapse in the use of biomass, which has also passed to district heating, as some biomass was combusted in cogeneration units that also produce green heat. As mentioned earlier, the auctions held in 2019 were the largest in terms of expected capacity to be installed in the European Union. The government expects that the implementation of all contracted capacity in these auctions and the new auctions planned for 2020 will deliver the 2020 target, although with a time delay of about 2 years.

*4.2. Summary of RES Auctions in 2016–2019*

According to the Energy Regulatory Office data, more than 2000 installations were supported by the auctions held in 2016–2019. As a result of the settlement of the conducted auctions, almost 154 TWh was sold in total, with expected generation within a period of 15 years from commissioning.

The total value of energy covered by the winning bids was over PLN 38 billion (EUR 8.84 billion). Most of the support—i.e., nearly PLN 37 billion (EUR 8.6 billion)—went towards new installations, i.e., those in which the generation of electricity will take place for the first time after the auction session closing. The existing installations (migrating from the system of green certificates) used from the pool allocated to them (worth over PLN 46 billion (EUR 10.7 billion)) just over PLN 1 billion (EUR

250 million), which represents slightly more than 2% of the total value of electricity allocated to date for sale by auction.

The main beneficiaries of the auction system are the operators and developers of onshore wind energy and PV. As a result of the outcome of the auctions of 2016–2019, approximately 3.4 GW of new capacity from wind power, approximately 1.7 GW from PV and less than 70 MW of new capacity from other RES technologies—biogas, hydro and biomass—may be developed in the near future. Figure 8 shows the installed capacity contracted in the auction scheme in the period 2016–2019.

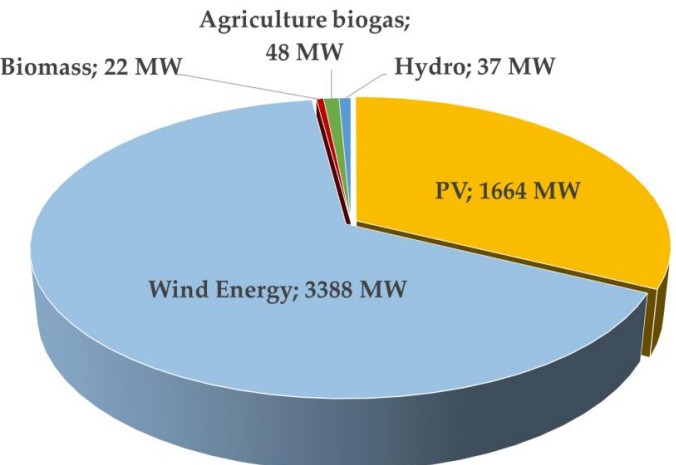

**Figure 8.** Expected installed capacity of RES units, resulting from the auctions conducted in the period 2016–2019 in Poland, in (MW).

The lowest average price from the winning bids was recorded in an auction dedicated to wind energy and amounted to less than PLN 200/MWh (2018 auction), which amounts to EUR 46/MWh. This was most likely due to the strong price pressure of this technology and the effect of the relatively small volume of electricity intended for sale at this dedicated auction, relative to market needs. This situation was further exacerbated by the fact that this was the first auction since the launch of the support system, i.e., since 2016, intended for new wind farms with an installed capacity of more than 1 MW. There was also a decrease in the cost of generating energy from photovoltaic installations: the average price from offers won in 2017 was PLN 372/MWh (EUR 86/MWh), in 2018 it was only PLN 352/MWh (EUR 82/MWh), and in 2019 it was PLN 317/MWh (EUR 74/MWh).

Currently we are observing the commissioning of the capacity resulting from the first auctions. This period—1–2 years—from the first auction was spent on further project development and the construction of units. At the end of 2019, a total of 406 new installations with a total capacity of 358.9 MW started generating electricity in the auction system. This means that, compared to the state at the end of 2018, 262 new sources with a total capacity of approximately 234.4 MW appeared last year. These are the units that won the auctions in previous years. Most of the new capacity is the result of the second auction of photovoltaic and wind units, in a basket of up to 1 MW, held in mid-2017. It resulted in the launch of 319 photovoltaic installations with a total capacity of 274.4 MW, and two wind units with a total capacity of 1.7 MW. According to information from the Energy Regulatory Office, the total number of new installations that started to produce energy in the auction system increased by 29 MW during the first quarter (Q1) of 2020. In total, by the end of Q1 2020, the total installed capacity of these new sources in the auction system increased to 384.146 MW.

The auction system is still in its beginning phase. The further development and construction of some technologies takes more than 2 years from the date of auction. Therefore, in order to fully evaluate the results of the auction, it is necessary to wait for more installations to produce electricity as scheduled.

## 5. Development Perspectives of Renewable Energy in Poland

### 5.1. National Energy and Climate Plan

The implementation of the Community's energy policy for 2020–2030, including in particular Regulation (EU) 2018/1999 of the European Parliament and the Council of 11 December 2018 on the Governance of the Energy Union and Climate Action, requires Member States to prepare and implement National Energy and Climate Plans [41]. On 30 December 2019, the Minister of State Assets sent the European Commission the Plan for Energy and Climate for Poland. The Plan presents assumptions and objectives, as well as policies and actions, for the implementation of the strategic directions of the energy union (energy security, internal energy market, energy efficiency, de-carbonization and research, innovation and competitiveness). Among the actions provided for in the Plan, also included are the plans for the development of renewable energy resources after 2020 in Poland [42].

The main objectives of Poland's energy and climate policy outlined in the document, and the means for providing a future measure of its implementation, are:

- Reduction of greenhouse gas emissions in sectors not covered by the European Union Emission Trading Scheme (EU ETS). This target was set at –7% in 2030 compared to 2005. The EU ETS foresees a 25% reduction in greenhouse gases between 2005 and 2030.
- As part of the EU-wide RES target for 2030, Poland declares to achieve a 21–23% share of energy being derived from renewable sources in the gross final consumption of energy (total consumption for electricity, heating and cooling and transport purposes) by 2030. It is estimated that by 2030, the share of RES in heating and cooling will increase by 1.1 percentage point on average annually. In transport, the share of renewable energy is expected to reach 14% by 2030.
- The national energy efficiency improvement target for 2030 has been set at 23% of primary energy consumption according to the PRIMES 2007 forecast, corresponding to the primary energy consumption of 91.3 Mtoe in 2030.

Figure 9 shows the expected $CO_2$ reductions in the EU ETS and non-ETS sector from 2005–2040. The plan assumes a significant $CO_2$ reduction in the energy sector—from about 380 million tons $CO_2$ emissions in 2020 to about 340 million tons $CO_2$ in 2030, and further to 270 million tons $CO_2$ in 2040.

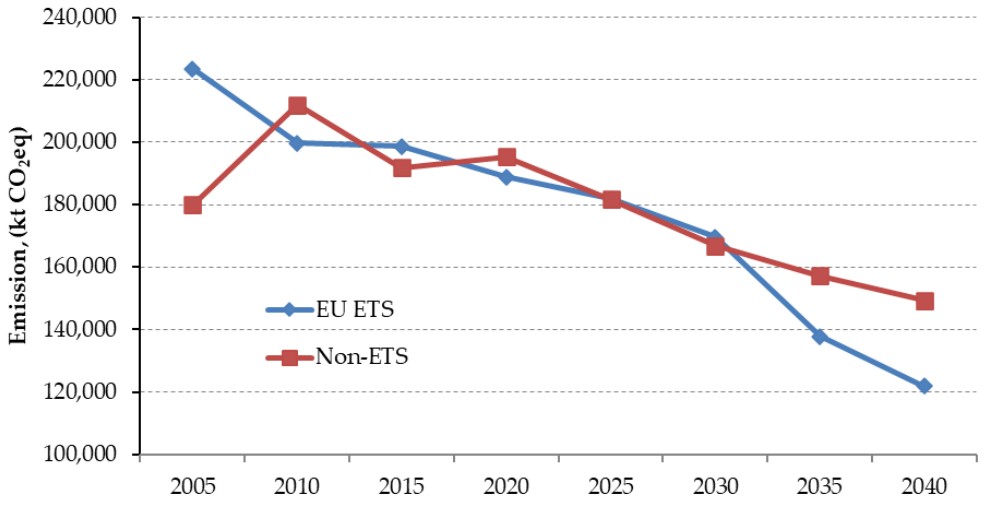

**Figure 9.** Expected emission in EU ETS and non- ETS sector in Poland [42].

In the National Energy and Climate Plan, the development of renewable energy sources has been presented as one of the tools for limiting greenhouse gas emissions. In particular, it was emphasized that achieving the 23% RES target will be possible if Poland is granted additional EU funds, including

the amount required for a fair transformation. The 23% RES target may be achieved via the indicative trajectory presented in Table 10.

**Table 10.** Proposal of trajectory of renewable energy according to the National Energy and Climate Plan [42].

|  | 2020 | 2022 | 2025 | 2027 | 2030 |
|---|---|---|---|---|---|
| Share of energy from RES in gross final energy consumption (%) | 15.0 | 16.4 | 18.4 | 20.2 | 23.0 |

The data used in the Plan for the implementation of this trajectory indicate the necessity of ensuring a significant increase of RES share in the electricity sector. Between 2015 and 2030, the share of RES in the power industry increases from 13.4% in 2015 to 31.8% in 2030. The development of RES will be controlled by the volume of energy announced by the Ministry through an auction. Technological progress will have a significant impact on the scale of RES application, both in terms of currently known methods of energy generation, but also in energy storage technologies. It is estimated that from the perspective of 2030, the share of RES in heating and cooling will increase by 1.1 percentage point on average annually, i.e., to the level of about 28.4%. In transport, the share of renewable energy is expected to reach 14% in 2030.

By 2030, the share of RES-derived electricity production will have increased from 13% in 2015 (23 TWh) by more more than double. The Plan predicts a 31.8% share and about 64 TWh of electricity generated from renewable sources in 2030. By 2040, it will reach about 40%, i.e., about 90 TWh, of which more than three quarters will be produced from wind units (about 55 TWh and 25% share in total production) and photovoltaic sources (about 15 TWh with 7% share in total generation of electricity). The net volume of electricity generated from RES in 2040 may even be four times higher than in 2015.

According to the Plan, most of the RES is used in district heating—both historically and in predictions. According to the projections presented in the Plan, the share of energy from derived RES in the heating and cooling sector will increase from 14.5% in 2015 to 28.4% in 2030. This means an increase by 13.9 percentage points between 2015 and 2030, and by 11 percentage points between 2020 and 2030 (the average annual increase is 1.1 percentage points, and this is in line with the recommended Renewable Energy Directive II). The analysis, which includes optimization of the whole national fuel and energy system, as well as the taking into account of the availability of primary energy carriers, has shown that the rate of development of RES in the Polish heating sector proposed in the RED II Directive can only be achieved if significant financial resources are committed. These funds will be used to completely rebuild the existing generation assets of heat-only boiler units, of which 97% are currently based on coal.

In the transport sector, the share of RES is expected to reach 14% in 2030. This objective is being implemented mainly based on the bio-components used in liquid fuels, as well as via increasing the use of electricity (especially in road transport). The development of biofuels from waste materials (mainly II generation) is also envisaged. Their quantity is conditioned by the limit of the content of first generation biofuels to a level not exceeding 7%. The achievement of the 2020 target in the transport sector will be very difficult. The Renewable Energy Directive offers additional expectation in relation to electromobility. This may constitute a substantial part of the renewable energy in the transport sector by the year 2030. Multipliers for renewable electricity supplied to the transport sector should be used for the promotion of renewable electricity in the transport sector and to reduce their comparative disadvantage in energy statistics. By use of multipliers, the share of biofuels and biogas used in transport may be considered to be twice its energy content; meanwhile the share of renewable electricity shall be considered to be four times its energy content when supplied to road vehicles, and may be considered to be 1.5 times its energy content when supplied to rail transport. With the

exception of fuels produced from food and feed crops, the share of fuels supplied in the aviation and maritime sectors shall be considered to be 1.2 times their energy content, as stated in the RES Directive.

Figure 10 shows the renewable energy utilization of each subsector and the share of renewable energy in their final energy consumptions.

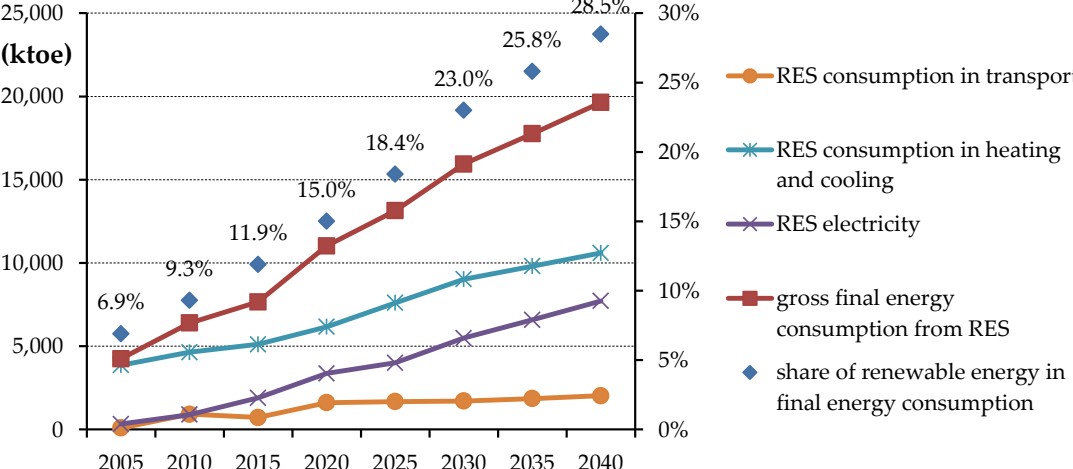

**Figure 10.** Energy from renewable energy sources by each subsector and the share of renewable energy in final energy consumption [42].

Based on the Union rules on State aid for energy, support for renewable energy sources should be based on competitive schemes that promote the reduction of costs associated with meeting targets. In Poland, the support for the development of renewable energy sources in the electricity sector is provided by the auction system described above, which is considered in the Plan as the most economically effective solution. The support granted as a result of the auction settlement offers stable conditions for investment in new RES installations. The adopted auction support system makes it possible to direct aid to selected areas and sectors, thus enabling the optimization of the energy mix in electricity. It has been indicated that the mechanisms of the support and promotion of energy generation from RES will be tailored to market needs (although this is expected to be necessary until about 2030), and will put forward solutions in a preferential way, thus ensuring maximum availability (high efficiency and coefficient of use, controlling, use of energy storage) with the relatively lowest cost of energy production, while meeting local energy demands (heat, electricity, transport) as well as those demands related to waste management (consistent with the hierarchy of waste management) and the use of local resources.

The plan has separated the role of and approach to individual RES technologies in the energy mix into steerable (controllable) and non-steerable (variable) sources.

The role of non-steerable RES is expected to be as follows:

- Solar energy (photovoltaics)—the advantage of this technology is a positive relationship between the intensity of sunlight and the daily demand for electricity, and an increased generation in summer correlated with the demand for cooling. These are installations of relatively small capacity, but the total installed capacity will be increasingly important for the national electricity system. The use of PV is an alternative to the use of post-industrial land and poor-quality agriculture land, as well as building roofs. They are of key importance for the current dynamic development of micro-installations, reinforced by dedicated financial support programs. It is estimated that photovoltaic sources will reach economic and technical maturity after 2022, i.e., they will not require operational support after that date. According to the projections of the National Energy and Climate Plan, the net maximum capacity of PV installations is expected to increase to about 7.3 GW in 2030, and about 16 GW in 2040.

- Offshore wind energy—the wind at sea reaches relatively high speeds and does not encounter obstacles (ground roughness and local social acceptance), which makes offshore wind power plants more productive than onshore ones. The commencement of investment in these capacities is determined by the completion of works on the reinforcement of the transmission grid in the northern part of the country, so that it is possible to transfer the power inland. The first offshore wind farm is expected to be integrated into the power system in 2025. The Polish coastline gives the possibility of implementing further offshore installations, but the possibility of balancing them in the national grid system will be crucial for the investment. It is expected that these sources will account for the largest amount of electricity generated from RES in 2040. According to the projections of the National Plan, the capacity of offshore wind is expected to increase to about 3.8 GW in 2030, and about 8 GW in 2040.

- Onshore wind energy—in the medium term, the increase n the share of this technology in the energy balance is expected to be less dynamic compared to previous years. A significant difficulty as presented in the Plan is the cost of balancing possibilities. Another problem is the varied level of acceptance of the construction of wind power plants by the local community. In order to reduce potential conflicts, the Plan supports investors in creating systems for the participation of residents in project implementation. According to the projections of the National Plan, it is expected that the onshore wind capacity will increase to about 9.6 GW in 2030, and maintain this volume until 2040.

- Hydropower—the Plan points to a negligible potential available for hydropower development. In the long term, the development of water energy may be influenced by the development of inland waterways and the revitalization of water dams, which are important from the point of view of river regulation and rational water management (preventing floods and droughts, increasing retention). It should be noted that the operation of run-of-river hydro power plants can be regulated, although to a limited extent. Pumped storage water plants are not classified as RES but perform a regulatory function for the grid system.

In terms of steerable sources, the Plan foresees a contribution:

- Energy from biomass (and heat from waste)—this source in Poland will be used in individual boilers in households for heating purposes, and in energy system mainly in cogeneration—it has the greatest potential for achieving the RES target in district heating due to the availability of fuel and the technical and economic parameters of the installation. Biomass-generating units should be located in the neighborhood of production (rural areas, wood industry regions, municipal waste disposal sites) and in places where it is possible to maximize the use of the primary energy contained in the fuel in order to minimize the environmental cost of transport. The use of biomass for energy also contributes to better waste management. It has been assumed that biomass-fired boilers will be a technology that will easily replace the power of coal-fired boilers in existing heat-only boiler units. A much wider use of biomass in heat generation than before (also in sectors such as municipal and commercial ones) is necessary in order to meet the requirements for increasing the share of RES in heat generation by at least 1.1 percentage point on average per year by 2030, on the one hand, and to contribute to the objective of achieving a 23% share of RES in gross final energy consumption, on the other. From the perspective of 2030, the consumption of biomass for heat production must increase almost 10-fold to 346 ktoe (in 2015, Poland used 36 ktoe of energy contained in solid biomass, while in 2016 and 2017 it was 58 and 66 ktoe, respectively).

- Energy from biogas—the use of biogas will be particularly useful in the cogeneration of electricity and heat. An advantage is the possibility of storing energy in biogas, which can be used for regulatory purposes. The main factor determining the small increase in the capacity installed in biogas plants so far is the high investment expenditure. However, from an economic point of view, biogas offers additional added value as it enables the utilization of particularly harmful waste (e.g., animal waste, landfill gas). Depending on local demand, biogas plants can supply electricity, biomethane and heating or cooling by using locally available resources.

- Geothermal energy—although its use is currently at a relatively low level, an upward trend is expected. The plan indicates that determining the geothermal potential requires large financial outlays with a high degree of uncertainty, but the use of this type of energy may determine the development of renewables.

*5.2. Electricity Production by Fuel in the National Energy and Climate Plan*

The Plan provides for gradual changes in the structure of electricity production, resulting from regulatory and market conditions. The development of renewable energy sources and the obligation to purchase $CO_2$ emission allowances under the ETS system will result in a gradual decrease in the share of coal-fired power plants in the electricity production structure [42].

The share of coal units in the production structure is expected to decrease from about 80% in 2015 to about 56% in 2030 (113 TWh). The main factor influencing this decrease in the share of coal is the shutdown of coal units and the decreasing operating time of old coal units. This is mainly due to the increase, as predicted in the Plan, in the use of low-emission sources, especially nuclear units, high-efficiency steam-gas units and the further increase in generation from RES units, especially offshore wind and photovoltaic. Nevertheless, despite the significant decrease in their share, coal-fired power plants will continue to be an important producer of electricity in the country, which is essential to ensuring uninterrupted energy supply to consumers.

Role of gas units: after 2024, the new units will be mainly high-efficiency steam-gas cogeneration units, as well as condensing units. Their share in domestic production will grow (from about 4% in 2015) two and a half times by 2030, after which, maintaining a similar rate of growth in subsequent years, it will increase to about 17% in 2040. The state's climate and energy policy will force the implementation of new low-emission sources, a large part of which will be non-steerable renewable sources characterized by the variability of production over time (wind and photovoltaic power plants). The existence of such unstable and low-carbon sources of generation in the projected quantities will require investment in flexible sources, demand side response (DSR), energy storage, etc. These sources will be essential for integration in the power system. For this reason, investments in gas units will be very important for the operation of power system. Gas units are flexible enough to meet the increased RES balancing requirements.

As mentioned earlier, the share of RES in electricity generation will be doubled by 2030. The plan assumes the generation of about 64 TWh of electricity from renewable sources by 2030, which will constitute a 31.8% share of electricity generated. By 2040, the production of electricity from RES sources is expected to reach about 90 TWh, which will constitute a 40% share of electricity generated. It is expected that more than three quarters will be produced from wind energy, both onshore and offshore (about 55 TWh), as well as photovoltaic (about 15 TWh).

Furthermore, an important element of the Plan's $CO_2$ reduction policy is the development of nuclear energy in Poland. The first nuclear power plant is expected to be operational by 2033, then two more will open in 2035 and 2037, and three more in intervals of 2–3 years. The estimated production from nuclear power plants in 2040 will amount to about 30.6 TWh, which constitutes a 14% share in total electricity production [42].

Figure 11 presents planned installed capacity in the Polish power system according to the National Energy and Climate Plan.

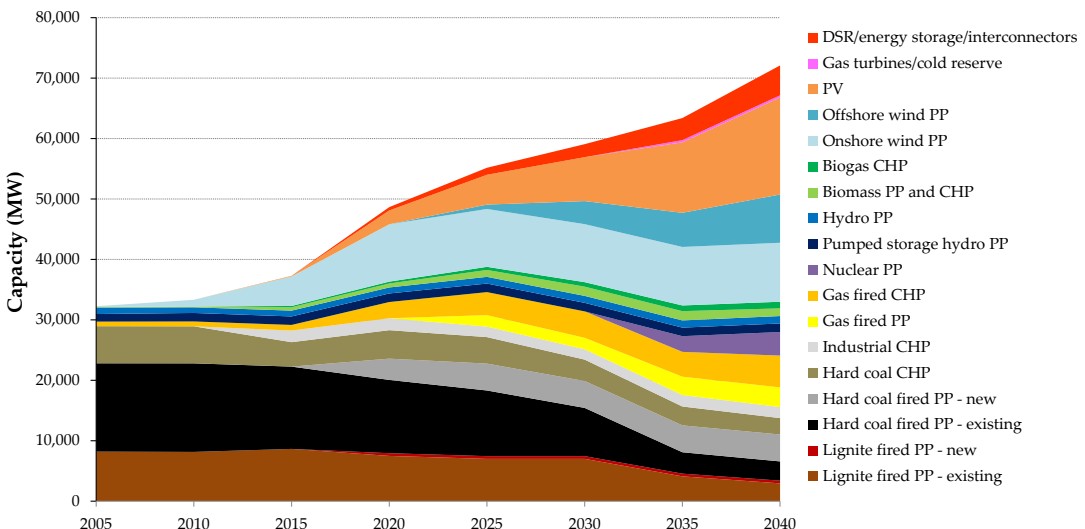

**Figure 11.** Installed capacity in Poland in the period 2005–2040 [42].

### 5.3. Forecast of Changes in Reference Prices of RES Auctions in 2020–2025

One of the fundamental questions that should be asked when analyzing the viability of RES power plants concerns the maximum price of energy sales that can be expected. Under the auction mechanism presented in Chapter 3, this price will be maximally equal to the reference price, i.e., a bid cannot be higher than the amount specified in the regulation set by the relevant Minister of Energy. For different power generation technologies these prices vary. Therefore, the authors have attempted to correlate these prices with the levelized cost of energy (LCOE). This is a parameter that reflects the ratio of the sum of total capital expenditure and operating costs incurred to the amount of energy produced in the same period [35,43]. It is described by (1).

$$LCOE = \frac{\sum_{t=0}^{n} \frac{I_t}{(1+r)^t} + \sum_{t=1}^{n} \frac{O\&M_t + F_t}{(1+r)^t}}{\sum_{t=1}^{n} \frac{E_t}{(1+r)^t}} \tag{1}$$

where $I_t$ is capital expenditure in year $t$ (CAPEX); $O\&M_t$ is operating cost in year $t$ (OPEX); $F_t$ is fuel cost in year $t$; $E_t$ is electricity produced in year $t$; $n$ is estimated lifetime of the power plant (construction and operation); and $r$ is discount rate.

The correlation study of the LCOE parameter with reference prices REF was performed using the Pearson correlation coefficient [44] (2). The aim of using this method is to indicate whether the Polish Ministry of Energy, while establishing reference prices at RES auctions, also considers changes in the value of the LCOE parameter. Unfortunately, due to the fact that the mechanism of RES support based on auction rules was only introduced in 2016, the quantity of data is limited. The authors are aware that more accurate results will be possible in the future; however, examining the reference price trends may provide an indication of the direction in which the Polish energy policy should go with regard to RES support.

$$r_{xy} = \frac{cov(x, y)}{\sigma_x \cdot \sigma_y} \tag{2}$$

Table 11 shows the values of LCOE [45], the RES auction reference prices for 2016–2018 and Pearson correlation coefficients.

**Table 11.** LCOE values (based on [45]) and reference prices (REF) at Polish RES auctions in 2016–2018, expressed in USD$_{2018}$/kWh.

| | | Wind Offshore | Wind Onshore < 1 MW | Wind Onshore > 1 MW | Solar < 1 MW | Solar > 1 MW | Hydro < 1 MW |
|---|---|---|---|---|---|---|---|
| **Year 2016** | REF, PLN/kWh | 0.47 | 0.3 | 0.385 | 0.465 | 0.445 | 0.48 |
| | REF, USD/kWh | 0.130 | 0.083 | 0.107 | 0.129 | 0.123 | 0.133 |
| | LCOE, USD/kWh | 0.132 | 0.065 | 0.065 | 0.119 | 0.119 | 0.053 |
| **Year 2017** | REF, PLN/kWh | 0.47 | 0.32 | 0.35 | 0.45 | 0.425 | 0.48 |
| | REF, USD/kWh | 0.130 | 0.089 | 0.097 | 0.125 | 0.118 | 0.133 |
| | LCOE, USD/kWh | 0.127 | 0.064 | 0.064 | 0.097 | 0.097 | 0.055 |
| **Year 2018** | REF, PLN/kWh | 0.45 | 0.31 | 0.35 | 0.42 | 0.4 | 0.5 |
| | REF, USD/kWh | 0.125 | 0.086 | 0.097 | 0.116 | 0.111 | 0.138 |
| | LCOE, USD/kWh | 0.126 | 0.055 | 0.055 | 0.085 | 0.085 | 0.048 |
| **Pearson correlation $r_{xy}$** | | **0.629** | **−0.091** | **0.577** | **0.936** | **0.973** | **−0.961** |

Table 11 shows that there is a positive, very strong correlation between the cost of photovoltaic generation and the reference prices at auction. The opposite is the case for the correlation between the aforementioned values for hydroelectric power plants.

Due to the interesting nature of the aforementioned results, the authors have verified, with the use of the linear regression method, at which level the reference prices at RES auctions and the LCOE parameter should be determined up till 2025. Figure 12 shows the results of a simulation performed in Microsoft Excel [46].

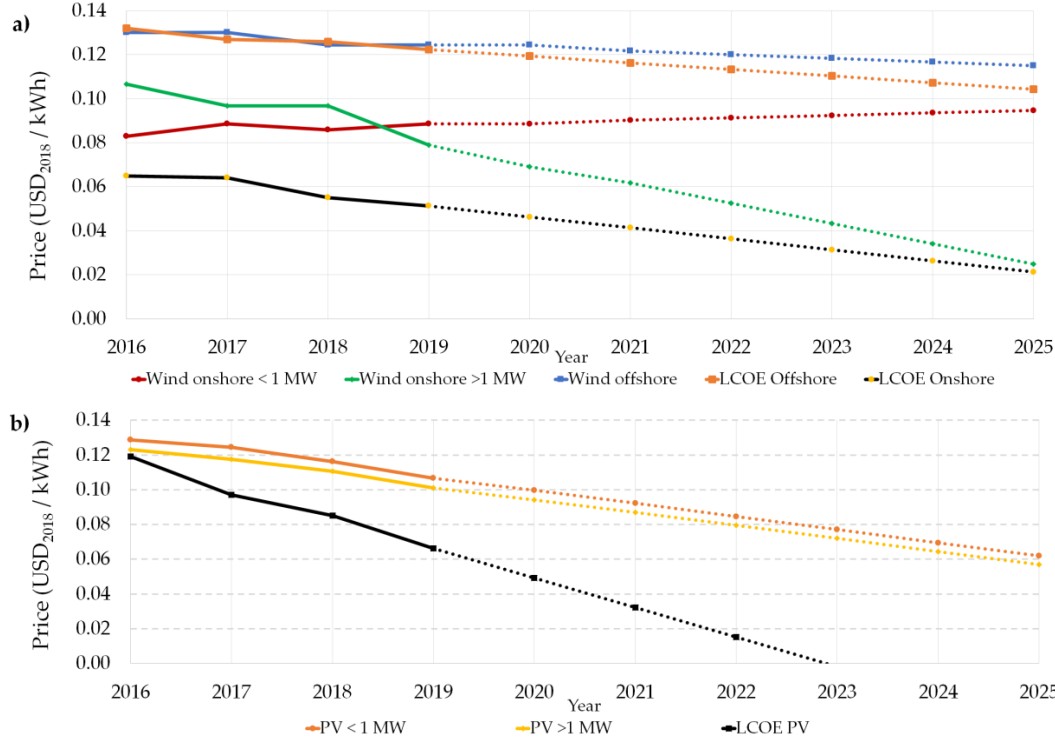

**Figure 12.** *Cont*.

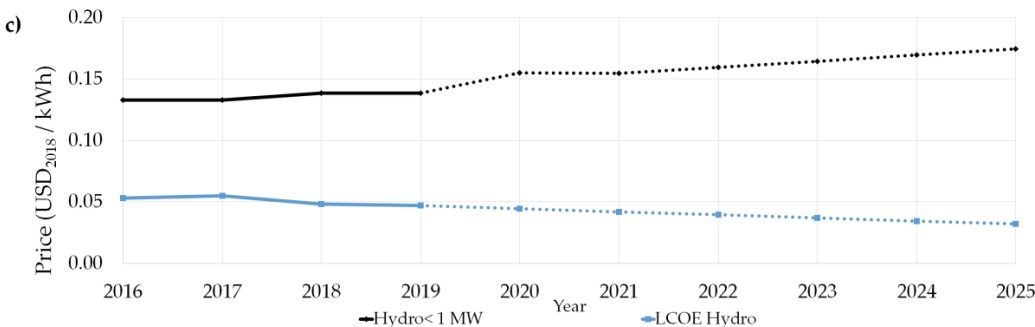

**Figure 12.** Forecast of reference prices at RES auctions in Poland and LCOE for (**a**) onshore and offshore wind farms, (**b**) PV power plants and (**c**) hydropower plants.

The following issues can be observed from Figure 12:

- Reference prices at RES auctions for onshore wind power plants with an installed capacity of less than 1 MW will continue to increase, while the levelized cost of energy will decrease. This is an obvious indication supporting the further development of wind power generation units below 1 MW.
- The LCOE value for offshore power plants will decrease faster than the planned decrease in reference prices at RES auctions. In order to encourage future investors to implement offshore projects, such a scenario seems to be quite likely.
- The projected LCOE value for PV power plants in 2024 is less than 0. This is an economically impossible case. Therefore, it can be assumed that the rate of decrease in the LCOE value for PV power plants in the coming years will be weaker than that resulting from linear regression. This forecast should therefore be revised in the future.
- The slope of curves representing reference price values at RES auctions for PV technology is less significant than that of the curve representing the LCOE parameter. This would provide a potential incentive for investors to further develop solar power plants.
- A similar situation can be observed for hydropower plants. According to forecasts, reference prices at RES auctions will increase while LCOE will decrease. In the case of obtaining a very high closing price, the investor could show higher profits.

Table 12 shows the Pearson correlation coefficients between forecasts of LCOE and reference prices at RES auctions in 2016–2025. It should be noted that they match the observations from Figure 12.

**Table 12.** Correlation of LCOE values and reference prices at Polish RES auctions in 2016–2025 (values for 2019–2025 based on linear regression forecast).

| Technology | Pearson Correlation $r_{xy}$ |
|---|---|
| Wind Offshore | 0.970 |
| Wind Onshore < 1 MW | −0.919 |
| Wind Onshore > 1 MW | 0.990 |
| Solar < 1 MW | 0.998 |
| Solar > 1 MW | 0.999 |
| Hydro < 1 MW | −0.976 |

Therefore, it can be concluded that the minister responsible for energy in Poland, while setting reference prices for RES auctions, is guided by the trends of changes in the LCOE values for solar power plants, wind power plants with installed capacity above 1 MW, and offshore wind power plants. In the

case of wind power plants with a capacity of less than 1 MW and small hydroelectric power plants, he gives an incentive to increase the interest in building sources in these energy generation technologies.

## 6. Conclusions and Discussion

The European Union's ambition is to derive 20% of the energy in the gross final consumption from renewable energy sources by 2020. This Community target has been distributed among the individual Member States, allocating to them a binding framework for the promotion of energy from renewable sources in Directive 2009/28/EC. Today, in 2020, it is time for settlements towards the achievement of these goals. Preliminary data show that some Member States had already successfully met their targets ahead of schedule, while some may be having difficulties in meeting them before the end of 2020. Furthermore, the European policy to promote the use of energy from renewable sources does not expire with the year 2020. Adopted in 2018, the revision of the Renewable Energy Sources Directive provides a new approach to the development of renewable sources beyond 2020. The new Directive set up a common objective at the European Union level with a 32% target for the share of energy derived from renewable sources in total EU energy consumption by 2030. This adopted target, together with the execution by Member States of National Energy and Climate Plans, will be a driver for renewable energy development in the next decade.

The development of renewable energy brings economic, environmental and social benefits. For a proper assessment, a full analysis of the benefits and costs of the development of renewable energy in Poland's specific environment is recommended, which may support political decisions and the preparation of new strategic documents for the development of the economy and energy sector.

There are currently a few main support mechanisms for deriving electricity from renewable energy sources in the EU, including investment subsidies, fixed price mechanisms in the forms of feed-in schemes or feed-in premiums, and quota system based on auctions or tradable green certificates. Poland decided in 2004 to choose a quota system based on tradable green certificates, and to implement an auction scheme as the system more preferred by the EU institutions, especially with adjusting the support to the new guidelines on state aid. It is expected that Poland will not achieve the RES target for 2020. Some delay in realization is expected. One of the reasons for delaying the achievement of this objective may be the long implementation process of the new auction support scheme, and the blocking of the development of some promising technologies such as wind power and the non-utilization of biomass potential have also contributed to the delay in achieving this goal. Nevertheless, the implementation of the Directives from 2001 and 2009 via Polish law has contributed to the increased use of energy from renewable sources in the EU Community. Only in the power industry has there been an increase from 2.8 TWh of electricity generated from RES in 2004 to approximately 24 TWh in 2019, and the installed capacity of RES has increased eight-fold, from 1150 MW in 2005 to more than 9000 MW now.

The Community and national policies on the promotion of energy from renewable sources are often disputed, and cause debates about the benefits and costs of developing renewable energy. This is due, inter alia, to the frequent divergence of private and public interests. There is no doubt that at the current level of development some technologies of renewable energy sources are still expensive, and without support, renewable energy would not be able to compete with energy generation units operating in the power-generation system. It is, however, not appropriate to conclude that the development of renewable energy is only associated with costs and an increase in the energy bill for end users [35,36,47].

There is still potential to reduce the costs of renewable energy source technologies. The costs of energy generation depend on local conditions and the potential of the renewable energy resource. We can observe that wind power plants are already being, and in the coming years PV will be, developed without support system, which are included in the Directives and then implemented via State legislation. Already now, the first projects are being constructed in Poland, thanks to power purchase agreements—so-called PPA. These power plants are being commissioned without the auction

support that has been implemented in Poland. These power plants will operate with income only from the sale of electricity.

Poland seems to be country with great potential for RES development. The Polish National Energy and Climate Plan, sent to the EU Commission, points to the ambitions of the current government in terms of the further development of renewable energy after 2020. The Plan predicts increases in the generation of electricity derived from RES from today's level of 24 TWh to about 64 TWh by 2030, and in the next decade to 90 TWh, which will constitute a 40% share of the electricity generated in 2040. Future decisions about the development of RES in the whole EU will have a strong impact on the entire Polish energy market.

**Author Contributions:** Conceptualization, J.P. and T.S.; methodology, J.P.; software, K.Z.; validation, J.P., T.S., and K.Z.; formal analysis, J.P.; investigation, K.Z.; resources, P.T.; data curation, P.T.; writing—original draft preparation, J.P., T.S., and K.Z.; writing—review and editing, J.P., T.S., and K.Z.; visualization, K.Z.; supervision, J.P. All authors have read and agreed to the published version of the manuscript.

**Funding:** This research received no external funding.

**Conflicts of Interest:** The authors declare no conflict of interest.

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
