# Peer review of "Electricity Generation from Renewable Energy Sources in Poland as a Part of Commitment to the Polish and EU Energy Policy"

_energies, doi:10.3390/en13164261_

Round 1

Reviewer 1 Report

See the attachment

Reviewer 2 Report

The authors present an interesting paper on the policy implementation in Poland to achieve the EU targets of renewable energy use in different sectors, with a focus on power generation. The paper is well written and clear, but it is more similar to a review paper rather than an original research article. The only original contribution of the authors is briefly presented in section 5.3, and I think it is too limited to justify a research paper.

So, my recommendation is to change the type of paper to review paper, possibly by including some additional references to academic papers dealing with similar subjects, maybe for other countries.

Additional remarks are reported below, in order of appearance in their manuscript:

Lines 76-77 (citing Figure 1): It is not clear why EU countries have such different targets compared to similar baselines. Please elaborate on this issue.

Lines 80-82: The COVID-19 pandemic will alter the energy consumption and renewable share in 2020, and in some cases a significant distortion may modify the results. A brief sentence acknowledging this issue may help the readers.

Table 1: The first column reports “2004”, but the previous paragraphs mentioned 2005 as baseline year. Please correct the values in the table.

Lines 116-120: Actually, the revised version of each National energy and climate plan was published in January for each country. The authors should mention this in their discussion.

Line 22: Actually, the reduction of the carbon footprint is the first objective of the use of RES in EU, not an additional benefit as suggested by the authors.

Lines 292-293: From the data presented in Table 3 it seems that PV plants will have basically no role in electricity generation. Are these numbers correct? In this case, please correct this sentence.

Figure 3: Are these values real numbers or rather policy targets? Could you include the real numbers as a comparison? I found them in the following sections, so maybe the authors could include a reference to the appropriate section to help the readers.

Lines 473-476: The authors should give an average value for the conversion of PLN to EUR or USD, which are commonly used for international electricity prices.

Figure 9: the image quality is rather low, please provide a better picture.

Line 705: This target also involves the use of multipliers (e.g. electricity in road transport is counted as four times its energy content, some advanced biofuels two times, etc.). This aspect should be mentioned for readers.

Section 5.3: Using the Pearson correlation coefficient on three annual values has no statistical significance, so I think the authors should choose a different approach, or precisely justify this choice by stating the normal uses of this coefficient and explain why they use it in this unusual situation.

Reviewer 3 Report

The paper presents energy scenarios regarding RES in Poland. All the information in support of the paper are recent and updated (data, laws). The paper is well written and interesting. Just dome minor corrections are required:

- The paper is rather lengthy. Since some parts can be easily summarized, I would suggest shortening the text a little.

- Figure 2: in the caption, you should display the year to whom stats are referred to

- row 298: "Offshore wind energy is assuming for implementation in 2027" Not clear. Maybe, "wind energy is assumED.."

- Avoid contracted forms in the written language (i.e. at row 310, write "DOES NOT")

- There are many figures and tables and it seems that information may be collapsed in fewer and 9 may be merged ones- I.e. (not mandatory, just to give an idea) Figure 4 information could be integrated in Table 1 / Table 8 and 9 may be merged. /Figures 12 13 and 14 could be merged under the same caption (i.e. Figure 12 a, b and c)

- Figure 5 and Figure 9: low quality, to be enhanced.

- Figure 5: it reports labels in Polish on the x-axis. Please provide labels in English. Prices are displayed with zloty as currency. It would be great to convert value into Euros to have a clear idea.

- Even in the body of the text and in Tables, energy prices are often expressed in the Polish currency.

- Provide a list of abbreviations used in the text

- revise dates format - ie. "April 1st, 2020" "January 1st,2016" "October 31st" "December 30th, 2019" and so on

- row 588: what is the ratio between fossil and biomass in co-firing? were there any limitation in accepting it as renewable plant, as stated in row 588?

- row 627: edit verb in passive form "this period was spenT"

- "13.9 percentage" and other-- just write %

- When you apply the definition of LCOE, do you refer strictly to the definition given by IEA? in your opinion, when it comes to renewable power sources, is there the need to revise the formula? There should be quite a lot literature to this end

- section 5.3 is very interesting. Could you elaborate a little more the method used for the calculation?

- rows 880-883: I agree with the point on the regression model. However, since it does not make any sense to display negative LCOE, in the plot adjust to zero Y-axis minimum value.

Is the any news about Poland from the most recent Green Deal that is of interest for the study?

Reviewer 4 Report

While the paper presents some interesting analysis and policy history for Poland, it requires a substantial edit.

The language is disjointed and there are glaring inconsistencies of terminology. For example, on page 1 the document utilises GW figures but also MW figures that are greater than 1 million. This could be simplified.

The analysis on LCOE and scheme outcomes is useful. As such, I would encourage the authors to do a thorough edit of the paper to improve its readability.

Round 2

Reviewer 2 Report

The authors have addressed my concerns.

Author Response

Thank you to the respected reviewer for his comments and suggestions. They significantly contributed to the improvement of the quality of the article. As for the language proofreading, we have made every effort to improve the English language of the last version of this article.

Reviewer 4 Report

Thanks to the authors for editing the text. I think it would benefit from another one or two rounds of editing.

I think the English is still a bit disjointed. In other words, I think the authors should carefully reread it to ensure it is easier to read.

Author Response

(The authors gave the same response as above.)
